# A Lyapunov-based Approach to Safe Reinforcement Learning

**Yinlam Chow**
DeepMind
yinlamchow@google.com

**Ofir Nachum**
Google Brain
ofirnachum@google.com

**Edgar Duenez-Guzman**
DeepMind
duenez@google.com

**Mohammad Ghavamzadeh**
Facebook AI Research
mgh@fb.com

## Abstract

In many real-world reinforcement learning (RL) problems, besides optimizing the main objective function, an agent must concurrently avoid violating a number of constraints. In particular, besides optimizing performance, it is crucial to guarantee the *safety* of an agent during training as well as deployment (e.g., a robot should avoid taking actions - exploratory or not - which irrevocably harm its hardware). To incorporate safety in RL, we derive algorithms under the framework of constrained Markov decision processes (CMDPs), an extension of the standard Markov decision processes (MDPs) augmented with constraints on expected cumulative costs. Our approach hinges on a novel *Lyapunov* method. We define and present a method for constructing Lyapunov functions, which provide an effective way to guarantee the global safety of a behavior policy during training via a set of local linear constraints. Leveraging these theoretical underpinnings, we show how to use the Lyapunov approach to systematically transform dynamic programming (DP) and RL algorithms into their safe counterparts. To illustrate their effectiveness, we evaluate these algorithms in several CMDP planning and decision-making tasks on a safety benchmark domain. Our results show that our proposed method significantly outperforms existing baselines in balancing constraint satisfaction and performance.

## 1 Introduction

Reinforcement learning (RL) has shown exceptional successes in a variety of domains such as video games [25] and recommender systems [40], where the main goal is to optimize a single return. However, in many real-world problems, besides optimizing the main objective (the return), there can exist several conflicting constraints that make RL challenging. In particular, besides optimizing performance it is crucial to guarantee the *safety* of an agent in deployment [5, 32, 33], as well as during training [2]. For example, a robot should avoid taking actions which irrevocably harm its hardware; a recommender system must avoid presenting harmful or offending items to users.

Sequential decision-making in non-deterministic environments has been extensively studied in the literature under the framework of Markov decision processes (MDPs). To incorporate safety into the RL process, we are particularly interested in deriving algorithms under the context of constrained Markov decision processes (CMDPs), which is an extension of MDPs with expected cumulative constraint costs. The additional constraint component of CMDPs increases flexibility in modeling problems with *trajectory-based* constraints, when compared with other approaches that customize immediate costs in MDPs to handle constraints [34]. As shown in numerous applications from robot motion planning [30, 26, 11], resource allocation [24, 18], and financial engineering [1, 41], it is more natural to define safety over the whole trajectory, instead of over particular state and action pairs. Under this framework, we denote an agent's behavior policy to be *safe* if it satisfies the cumulative cost constraints of the CMDP.

Despite the capabilities of CMDPs, they have not been very popular in RL. One main reason is that, although optimal policies of finite CMDPs are Markov and stationary, and with known models the CMDP can be solved using linear programming (LP) [3], it is unclear how to extend this algorithm to handle cases when the model is unknown, or when the state and action spaces are large or continuous. A well-known approach to solve CMDPs is the Lagrangian method [4, 15], which augments the standard expected reward objective with a penalty on constraint violation. With a fixed Lagrange multiplier, one can use standard dynamic programming (DP) or RL algorithms to solve for an optimal policy. With a learnable Lagrange multiplier, one must solve the resulting saddle point problem. However, several studies [21] showed that iteratively solving the saddle point is apt to run into numerical stability issues. More importantly, the Lagrangian policy is only safe *asymptotically* and makes little guarantee with regards to safety of the behavior policy during each training iteration.

Motivated by these observations, several recent works have derived surrogate algorithms for solving CMDPs, which transform the original constraint to a more conservative one that yields an easier problem to solve. A straight-forward approach is to replace the cumulative constraint cost with a conservative *stepwise* surrogate constraint [9] that only depends on the current state-action pair. Since this surrogate constraint can be easily embedded into the admissible control set, this formulation can be modeled by an MDP that has a restricted set of admissible actions. Another surrogate algorithm was proposed by [14] in which the algorithm first computes a uniform *super-martingale* constraint value function surrogate w.r.t. all policies, and then finds a CMDP feasible policy by optimizing the surrogate problem using the *lexicographical ordering* method [39]. These methods are advantageous in the sense that (i) there are RL algorithms available to handle the surrogate problems (for example see [12] for the step-wise surrogate and [27] for the super-martingale surrogate), (ii) the policy returned by this method is safe, even during training. However, the main drawback of these approaches is their conservativeness. Characterizing sub-optimality performance of the corresponding solution policy also remains a challenging task. On the other hand, recently in policy gradient, [2] proposed the constrained policy optimization (CPO) method that extends trust-region policy optimization (TRPO) to handle the CMDP constraints. While this algorithm is scalable and its policy is safe during training, applying this methodology to more general RL algorithms (that are not in the family of proximal PG algorithms) is quite non-trivial.

Lyapunov functions have been extensively used in control theory to analyze the stability of dynamic systems [20, 28]. A Lyapunov function is a type of scalar potential function that keeps track of the *energy* that a system continually dissipates. Besides modeling physical energy, Lyapunov functions can also represent abstract quantities, such as the steady-state performance of a Markov process [16]. In many fields, Lyapunov functions provide a powerful paradigm to translate global properties of a system to local ones and vice-versa. Using Lyapunov functions in RL was first studied by [31], where Lyapunov functions were used to guarantee closed-loop stability of an agent. Recently [6] used Lyapunov functions to guarantee a model-based RL agent's ability to re-enter an "attraction region" during exploration. However, no previous works have used Lyapunov approaches to explicitly model constraints in a CMDP. Furthermore, one major drawback of these approaches is that the Lyapunov functions are hand-crafted, and there are no principled guidelines on designing Lyapunov functions that can guarantee the agent's performance.

The contribution of this paper is four-fold. First, we formulate the problem of safe RL as a CMDP and propose a novel *Lyapunov approach* to solve it. While the main challenge of other Lyapunov-based methods is to design a Lyapunov function candidate, we propose an LP-based algorithm to construct Lyapunov functions w.r.t. generic CMDP constraints. We also show that our method is guaranteed to always return a feasible policy, and under certain technical assumptions, it achieves optimality. Second, leveraging the theoretical underpinnings of the Lyapunov approach, we present two safe DP algorithms – safe policy iteration (SPI) and safe value iteration (SVI) – and analyze the feasibility and performance of these algorithms. Third, to handle unknown environments and large state/action spaces, we develop two scalable safe RL algorithms – (i) *safe DQN*, an off-policy fitted $Q$-iteration method, and (ii) *safe DPI*, an approximate policy iteration method. Fourth, to illustrate the effectiveness of these algorithms, we evaluate them in several tasks on a benchmark 2D planning problem and show that they outperform common baselines in terms of balancing performance and constraint satisfaction.

## 2 Preliminaries

We consider RL problems in which the agent's interaction with the system is modeled as a Markov decision process (MDP). A MDP is a tuple $(\mathcal{X}, \mathcal{A}, c, P, x_0)$, where $\mathcal{X} = \mathcal{X}' \cup \{x_{\text{Term}}\}$ is the state space, with transient state space $\mathcal{X}'$ and terminal state $x_{\text{Term}}$; $\mathcal{A}$ is the action space;

$c(x, a) \in [0, C_{\max}]$ is the immediate cost function (negative reward); $P(\cdot|x, a)$ is the transition probability distribution; and $x_0 \in \mathcal{X}'$ is the initial state. Our results easily generalize to random initial states and random costs, but for simplicity we will focus on the case of deterministic initial state and immediate cost. In a more general setting where cumulative constraints are taken into account, we define a constrained Markov decision process (CMDP), which extends the MDP model by introducing additional costs and associated constraints. A CMDP is defined by $(\mathcal{X}, \mathcal{A}, c, d, P, x_0, d_0)$, where the components $\mathcal{X}, \mathcal{A}, c, P, x_0$ are the same for the unconstrained MDP; $d(x) \in [0, D_{\max}]$ is the immediate constraint cost; and $d_0 \in \mathbb{R}_{\geq 0}$ is an upper-bound on the expected cumulative (through time) constraint cost. To formalize the optimization problem associated with CMDPs, let $\Delta$ be the set of Markov stationary policies, i.e., $\Delta(x) = \{\pi(\cdot|x) : \mathcal{X} \to \mathbb{R}_{\geq 0s} : \sum_a \pi(a|x) = 1\}$, for any state $x \in \mathcal{X}$. Also let $\mathrm{T}^*$ be a random variable corresponding to the first-hitting time of the terminal state $x_{\text{Term}}$ induced by policy $\pi$. In this paper, we follow the standard notion of transient MDPs and assume that the first-hitting time is uniformly bounded by an upper bound $\overline{\mathrm{T}}$ for any stationary policies [10]. This assumption implies that every stationary policy is *proper* [7], whose induced Markov chain has an *absorbing* property (see [13] for an example). While this assumption may seem restrictive, it is a standard one in *stochastic shortest path* problems for showing that the Bellman operator is a contraction. Its justification follows from the fact that sample trajectories collected in most RL algorithms consist of a finite stopping time (also known as a time-out); In general this assumption may also be relaxed in cases where a discount factor $\gamma < 1$ is applied on future costs. For notational convenience, at each state $x \in \mathcal{X}'$, we define the generic Bellman operator w.r.t. policy $\pi \in \Delta$ and generic cost function $h$: $T_{\pi,h}[V](x) = \sum_a \pi(a|x)\left[h(x, a) + \sum_{x' \in \mathcal{X}} P(x'|x, a)V(x')\right]$.

Given a policy $\pi \in \Delta$, an initial state $x_0$, the cost function is defined as $\mathcal{C}_\pi(x_0) := \mathbb{E}\left[\sum_{t=0}^{\mathrm{T}^*-1} c(x_t, a_t) \mid x_0, \pi\right]$, and the safety constraint is defined as $\mathcal{D}_\pi(x_0) \leq d_0$, where the safety constraint function is given by $\mathcal{D}_\pi(x_0) := \mathbb{E}\left[\sum_{t=0}^{\mathrm{T}^*-1} d(x_t) \mid x_0, \pi\right]$. In general the CMDP problem we wish to solve is given as follows:

> **Problem** $\mathcal{OPT}$**:** Given an initial state $x_0$ and a threshold $d_0$, solve $\min_{\pi \in \Delta} \left\{\mathcal{C}_\pi(x_0) : \mathcal{D}_\pi(x_0) \leq d_0\right\}$. If there is a non-empty solution, the optimal policy is denoted by $\pi^*$.

Under the transient CMDP assumption, Theorem 8.1 in [3] shows that if the feasibility set is non-empty, then there exists an optimal policy in the class of stationary Markovian policies $\Delta$. To motivate the CMDP formulation studied in this paper, in Appendix A, we include two real-world examples in modeling safety using (i) the reachability constraint, and (ii) the constraint that limits the agent's visits to undesirable states. Recently there has been a number of works on CMDP algorithms; their details can be found in Appendix B.

## 3  A Lyapunov Approach to Solve CMDPs

In this section, we develop a novel methodology for solving CMDPs using the *Lyapunov approach*. To start with, without loss of generality we assume to have access to a *baseline* feasible policy of the $\mathcal{OPT}$ problem, namely $\pi_B \in \Delta$.[1] We define a non-empty[2] set of Lyapunov functions w.r.t. the initial state $x_0 \in \mathcal{X}$ and constraint threshold $d_0$ as

$$\mathcal{L}_{\pi_B}(x_0, d_0) = \left\{L : \mathcal{X} \to \mathbb{R}_{\geq 0} : T_{\pi_B, d}[L](x) \leq L(x), \forall x \in \mathcal{X}'; \ L(x) = 0, \ \forall x \in \mathcal{X} \backslash \mathcal{X}'; \ L(x_0) \leq d_0\right\}. \tag{1}$$

For any arbitrary Lyapunov function $L \in \mathcal{L}_{\pi_B}(x_0, d_0)$, we denote by $\mathcal{F}_L(x) = \left\{\pi(\cdot|x) \in \Delta : T_{\pi, d}[L](x) \leq L(x)\right\}$ the set of $L-$induced Markov stationary policies. Since $T_{\pi, d}$ is a contraction mapping [7], any $L-$induced policy $\pi$ has the following property: $\mathcal{D}_\pi(x) = \lim_{k \to \infty} T_{\pi, d}^k[L](x) \leq L(x), \forall x \in \mathcal{X}'$. Together with the property of $L(x_0) \leq d_0$, this further implies any $L-$induced policy is a feasible policy of the $\mathcal{OPT}$ problem. However, in general the set $\mathcal{F}_L(x)$ does not necessarily contain any optimal policies of the $\mathcal{OPT}$ problem , and our main contribution is to design a Lyapunov function (w.r.t. a baseline policy) that provides this guarantee. In other words, our main goal is to construct a Lyapunov function $L \in \mathcal{L}_{\pi_B}(x_0, d_0)$ such that

$$L(x) \geq T_{\pi^*, d}[L](x), \ L(x_0) \leq d_0. \tag{2}$$

Before getting into the main results, we consider the following important technical lemma, which states that with appropriate *cost-shaping*, one can always transform the constraint value function $\mathcal{D}_{\pi^*}(x)$ w.r.t. optimal policy $\pi^*$ into a Lyapunov function that is induced by $\pi_B$, i.e., $L_\epsilon(x) \in \mathcal{L}_{\pi_B}(x_0, d_0)$. The proof of this lemma can be found in Appendix C.1.

**Lemma 1.** *There exists an auxiliary constraint cost* $\epsilon : \mathcal{X}' \to \mathbb{R}$ *such that a Lyapunov function is given by* $L_\epsilon(x) = \mathbb{E}\left[\sum_{t=0}^{\mathrm{T}^*-1} d(x_t) + \epsilon(x_t) \mid \pi_B, x\right]$, $\forall x \in \mathcal{X}'$, *and* $L_\epsilon(x) = 0$, $\forall x \in \mathcal{X} \setminus \mathcal{X}'$. *Moreover,* $L_\epsilon$ *is equal to the constraint value function w.r.t.* $\pi^*$, *i.e.,* $L_\epsilon(x) = \mathcal{D}_{\pi^*}(x)$.

From the structure of $L_\epsilon$, one can see that the auxiliary constraint cost function $\epsilon$ is uniformly bounded by $\epsilon^*(x) := 2\overline{\mathrm{T}} D_{\max} D_{TV}(\pi^*||\pi_B)(x),$[3] i.e., $\epsilon(x) \in [-\epsilon^*(x), \epsilon^*(x)]$, for any $x \in \mathcal{X}'$. However, in general it is unclear how to construct such a cost-shaping term $\epsilon$ without explicitly knowing $\pi^*$ a-priori. Rather, inspired by this result, we consider the bound $\epsilon^*$ to propose a Lyapunov function *candidate* $L_{\epsilon^*}$. Immediately from its definition, this function has the following properties:

$$L_{\epsilon^*}(x) \geq T_{\pi_B, d}[L_{\epsilon^*}](x), \ L_{\epsilon^*}(x) \geq \max\left\{\mathcal{D}_{\pi^*}(x), \mathcal{D}_{\pi_B}(x)\right\} \geq 0, \forall x \in \mathcal{X}'. \tag{3}$$

The first property is due to the facts that: (i) $\epsilon^*$ is a non-negative cost function; (ii) $T_{\pi_B, d+\epsilon^*}$ is a contraction mapping, which by the fixed point theorem [7] implies $L_{\epsilon^*}(x) = T_{\pi_B, d+\epsilon^*}[L_{\epsilon^*}](x) \geq T_{\pi_B, d}[L_{\epsilon^*}](x), \forall x \in \mathcal{X}'$. For the second property, from the above inequality one concludes that the Lyapunov function $L_{\epsilon^*}$ is a uniform upper-bound to the constraint cost, i.e., $L_{\epsilon^*}(x) \geq \mathcal{D}_{\pi_B}(x)$, because the constraint cost $\mathcal{D}_{\pi_B}(x)$ w.r.t. policy $\pi_B$ is the unique solution to the fixed-point equation $T_{\pi_B, d}[V](x) = V(x), x \in \mathcal{X}'$. On the other hand, by construction, $\epsilon^*(x)$ is an upper-bound of the cost-shaping term $\epsilon(x)$. Therefore, Lemma 1 implies that the Lyapunov function $L_{\epsilon^*}$ is a uniform upper-bound to the constraint cost w.r.t. optimal policy $\pi^*$, i.e., $L_{\epsilon^*}(x) \geq \mathcal{D}_{\pi^*}(x)$.

To show that $L_{\epsilon^*}$ is a Lyapunov function that satisfies (2), we propose the following condition that enforces a baseline policy $\pi_B$ to be *sufficiently close* to an optimal policy $\pi^*$.

**Assumption 1.** *The feasible baseline policy* $\pi_B$ *satisfies the condition* $\max_{x \in \mathcal{X}'} \epsilon^*(x) \leq D_{\max} \cdot \min\left\{\frac{d_0 - \mathcal{D}_{\pi_B}(x_0)}{\overline{\mathrm{T}} D_{\max}}, \frac{\overline{\mathrm{T}} D_{\max} - \overline{\mathcal{D}}}{\overline{\mathrm{T}} D_{\max} + \overline{\mathcal{D}}}\right\}$, *where* $\overline{\mathcal{D}} = \max_{x \in \mathcal{X}'} \max_\pi \mathcal{D}_\pi(x)$.

This condition characterizes the maximum allowable distance between $\pi_B$ and $\pi^*$, such that the set of $L_{\epsilon^*}-$induced policies contains an optimal policy. To formalize this claim, we have the following main result showing that $L_{\epsilon^*} \in \mathcal{L}_{\pi_B}(x_0, d_0)$, and the set of policies $\mathcal{F}_{L_{\epsilon^*}}$ contains an optimal policy.

**Theorem 1.** *Suppose the baseline policy* $\pi_B$ *satisfies Assumption* 1, *then on top of the properties in* (3), *the Lyapunov function candidate* $L_{\epsilon^*}$ *also satisfies the properties in* (2), *and thus, its induced feasible set of policies* $\mathcal{F}_{L_{\epsilon^*}}$ *contains an optimal policy.*

The proof of this theorem is given in Appendix C.2. Suppose the distance between the baseline and optimal policies can be estimated effectively. Using the above result, one can immediately determine if the set of $L_{\epsilon^*}-$induced policies contain an optimal policy. Equipped with the set of $L_{\epsilon^*}-$induced feasible policies, consider the following *safe* Bellman operator:

$$T[V](x) = \begin{cases} \min_{\pi \in \mathcal{F}_{L_{\epsilon^*}}(x)} T_{\pi, c}[V](x) & \text{if } x \in \mathcal{X}' \\ 0 & \text{otherwise} \end{cases}. \tag{4}$$

Using standard analysis of Bellman operators, one can show that $T$ is a monotonic and contraction operator (see Appendix C.3 for the proof). This further implies that the solution of the fixed-point equation $T[V](x) = V(x), \forall x \in \mathcal{X}$ is unique. Let $V^*$ be such a value function. The following theorem shows that under Assumption 1, $V^*(x_0)$ is a solution to the $\mathcal{OPT}$ problem.

**Theorem 2.** *Suppose that the baseline policy* $\pi_B$ *satisfies Assumption 1. Then, the fixed-point solution at* $x = x_0$, *i.e.,* $V^*(x_0)$, *is equal to the solution of the* $\mathcal{OPT}$ *problem. Furthermore, an optimal policy can be constructed by* $\pi^*(\cdot|x) \in \arg\min_{\pi \in \mathcal{F}_{L_{\epsilon^*}}(x)} T_{\pi, c}[V^*](x)$, $\forall x \in \mathcal{X}'$.

The proof of this theorem can be found in Appendix C.4. This shows that under Assumption 1, an optimal policy of the $\mathcal{OPT}$ problem can be found using standard DP algorithms. Note that verifying whether $\pi_B$ satisfies this assumption is still challenging, because one requires a good estimate of $D_{TV}(\pi^*||\pi_B)$. Yet to the best of our knowledge, this is *the first result* that connects the optimality of CMDP to Bellman's principle of optimality. Another key observation is that in

practice, we will explore ways of approximating $\epsilon^*$ via *bootstrapping* and empirically show that this approach achieves good performance, while guaranteeing safety at each iteration. In particular, in the next section, we will illustrate how to systematically construct a Lyapunov function using an LP in both planning and RL (when the model is unknown and/or we use function approximation) scenarios in order to guarantee safety during learning.

## 4 Safe Reinforcement Learning Using Lyapunov Functions

Motivated by the challenge of computing a Lyapunov function $L_{\epsilon^*}$ such that its induced set of policies contains $\pi^*$, in this section, we approximate $\epsilon^*$ with an auxiliary constraint cost $\widetilde{\epsilon}$, which is the *largest* auxiliary cost that satisfies the Lyapunov condition: $L_{\widetilde{\epsilon}}(x) \geq T_{\pi_B,d}[L_{\widetilde{\epsilon}}](x)$, $\forall x \in \mathcal{X}'$, and the safety condition $L_{\widetilde{\epsilon}}(x_0) \leq d_0$. The larger the $\widetilde{\epsilon}$, the larger the set of policies $\mathcal{F}_{L_{\widetilde{\epsilon}}}$. Thus, by choosing the largest such auxiliary cost, we hope to have a better chance of including the optimal policy $\pi^*$ in the set of feasible policies. So, we consider the following LP problem:

$$\widetilde{\epsilon} \in \arg\max_{\epsilon:\mathcal{X}' \to \mathbb{R}_{\geq 0}} \left\{ \sum_{x \in \mathcal{X}'} \epsilon(x) : d_0 - \mathcal{D}_{\pi_B}(x_0) \geq \mathbf{1}(x_0)^\top (I - \{P(x'|x,\pi_B)\}_{x,x' \in \mathcal{X}'})^{-1}\epsilon \right\}. \quad (5)$$

Here $\mathbf{1}(x_0)$ represents a one-hot vector in which the non-zero element is located at $x = x_0$.

On the other hand, whenever $\pi_B$ is a feasible policy, the problem in (5) always has a non-empty solution.[4] Furthermore, note that $\mathbf{1}(x_0)^\top (I - \{P(x'|x,\pi_B)\}_{x,x' \in \mathcal{X}'})^{-1}\mathbf{1}(x)$ represents the total visiting probability $\mathbb{E}[\sum_{t=0}^{T^*-1} \mathbf{1}\{x_t = x\} \mid x_0, \pi_B]$ from the initial state $x_0$ to any state $x \in \mathcal{X}'$, which is a non-negative quantity. Therefore, using the extreme point argument in LP [23], one can simply conclude that the maximizer of problem (5) is an indicator function whose non-zero element locates at state $\underline{x}$ that corresponds to the minimum total visiting probability from $x_0$, i.e., $\widetilde{\epsilon}(x) = (d_0 - \mathcal{D}_{\pi_B}(x_0)) \cdot \mathbf{1}\{x = \underline{x}\}/\mathbb{E}[\sum_{t=0}^{T^*-1} \mathbf{1}\{x_t = \underline{x}\} \mid x_0, \pi_B] \geq 0, \forall x \in \mathcal{X}'$, where $\underline{x} \in \arg\min_{x \in \mathcal{X}'} \mathbb{E}[\sum_{t=0}^{T^*-1} \mathbf{1}\{x_t = x\} \mid x_0, \pi_B]$. On the other hand, suppose that we further restrict the structure of $\widetilde{\epsilon}(x)$ to be a constant function, i.e., $\widetilde{\epsilon}(x) = \widetilde{\epsilon}$, $\forall x \in \mathcal{X}'$. Then, one can show that the maximizer is given by $\widetilde{\epsilon}(x) = (d_0 - \mathcal{D}_{\pi_B}(x_0))/\mathbb{E}[T^* \mid x_0, \pi_B]$, $\forall x \in \mathcal{X}'$, where $\mathbf{1}(x_0)^\top (I - \{P(x'|x,\pi_B)\}_{x,x' \in \mathcal{X}'})^{-1}[1, \ldots, 1]^\top = \mathbb{E}[T^* \mid x_0, \pi_B]$ is the expected stopping time of the transient MDP. In cases where computing the expected stopping time is expensive, one reasonable approximation is to replace the denominator of $\widetilde{\epsilon}$ with the upper-bound $\overline{T}$.

Using this Lyapunov function $L_{\widetilde{\epsilon}}$, we propose the safe policy iteration (SPI) in Algorithm 1, in which the Lyapunov function is updated via *bootstrapping*, i.e., at each iteration $L_{\widetilde{\epsilon}}$ is recomputed using (5), w.r.t. the current baseline policy. Properties of SPI are summarized in the following proposition.

---

**Algorithm 1** Safe Policy Iteration (SPI)

---

**Input:** Initial feasible policy $\pi_0$;
**for** $k = 0, 1, 2, \ldots$ **do**
    **Step 0:** With $\pi_b = \pi_k$, evaluate the Lyapunov function $L_{\epsilon_k}$, where $\epsilon_k$ is a solution of (5)
    **Step 1:** Evaluate the cost value function $V_{\pi_k}(x) = \mathcal{C}_{\pi_k}(x)$
    **Step 2:** Update the policy by solving the problem $\pi_{k+1}(\cdot|x) \in \arg\min_{\pi \in \mathcal{F}_{L_{\epsilon_k}}(x)} T_{\pi,c}[V_{\pi_k}](x), \forall x \in \mathcal{X}'$
**end for**
**Return** Final policy $\pi_{k^*}$

---

**Proposition 1.** *Algorithm 1 has the following properties:* (i) Consistent Feasibility*, i.e., suppose that the current policy $\pi_k$ is feasible, then the updated policy $\pi_{k+1}$ is also feasible, i.e., $\mathcal{D}_{\pi_k}(x_0) \leq d_0$ implies $\mathcal{D}_{\pi_{k+1}}(x_0) \leq d_0$; (ii) Monotonic Policy Improvement, i.e., the cumulative cost induced by $\pi_{k+1}$ is lower than or equal to that by $\pi_k$, i.e., $\mathcal{C}_{\pi_{k+1}}(x) \leq \mathcal{C}_{\pi_k}(x)$, $\forall x \in \mathcal{X}'$; (iii) Convergence, i.e., if we add a strictly concave regularizer to the optimization problem* (5) *and a strictly convex regularizer to the policy optimization step, then the policy sequence asymptotically converges.*[5]

The proof of this proposition is given in Appendix C.5, and the sub-optimality performance bound of SPI can be found in Appendix C.6. Analogous to SPI, we also propose a safe value iteration (SVI), in which the Lyapunov function estimate is updated at every iteration via bootstrapping, using the current optimal value estimate. Details of SVI is given in Algorithm 2 and its properties are summarized in the following proposition, whose proof is given in Appendix C.7.

**Proposition 2.** *Algorithm 2 has: (i) Consistent Feasibility and (ii) Convergence.*

To justify the notion of bootstrapping in both SVI and SPI, the Lyapunov function is updated based on the *best* baseline policy (the policy that is feasible and by far has the lowest cumulative cost). Once the current baseline policy $\pi_k$ is *sufficiently close* to an optimal policy $\pi^*$, then by Theorem 1, one may conclude that the $L_{\widetilde{\epsilon}}-$induced set of policies contains an optimal policy. Although these algorithms do not have optimality guarantees, empirically, they often return a near-optimal policy.

At each iteration, the policy optimization step in SPI and SVI requires solving $|\mathcal{X}'|$ LP sub-problems, where each of them has $|\mathcal{A}| + 2$ constraints and has a $|\mathcal{A}|-$dimensional decision-variable. Collectively, at each iteration its complexity is $O(|\mathcal{X}'||\mathcal{A}|^2(|\mathcal{A}| + 2))$. While in the worst case SVI converges in $K = O(\overline{\mathrm{T}})$ steps [7] and SPI converges in $K = O(|\mathcal{X}'||\mathcal{A}|\overline{\mathrm{T}}\log\overline{\mathrm{T}})$ steps [38], in practice, $K$ is much smaller than $|\mathcal{X}'||\mathcal{A}|$. Therefore, even with the additional complexity of policy evaluation in SPI that is $O(\overline{\mathrm{T}}|\mathcal{X}'|^2)$, or the complexity of updating $Q-$function in SVI that is $O(|\mathcal{A}|^2|\mathcal{X}'|^2)$, the complexity of these methods is $O(K|\mathcal{X}'||\mathcal{A}|^3 + K|\mathcal{X}'|^2|\mathcal{A}|^2)$, which in practice is much lower than that of the dual LP method, whose complexity is $O(|\mathcal{X}'|^3|\mathcal{A}|^3)$ (see Appendix B for more details).

---

**Algorithm 2** Safe Value Iteration (SVI)

---

**Input:** Initial $Q$-function $Q_0$; Initial Lyapunov function $L_{\epsilon_0}$ w.r.t. auxiliary cost function $\epsilon_0(x) = 0$;
**for** $k = 0, 1, 2, \ldots$ **do**
    **Step 0:** Compute $Q$-function $Q_{k+1}(x, a) = c(x, a) + \sum_{x'} P(x'|x, a) \min_{\pi \in \mathcal{F}_{L_{\epsilon_k}}(x')} \pi(\cdot|x')^\top Q_k(x', \cdot)$
    and policy $\pi_k(\cdot|x) \in \arg\min_{\pi \in \mathcal{F}_{L_{\epsilon_k}}(x)} \pi(\cdot|x)^\top Q_k(x, \cdot)$
    **Step 1:** With $\pi_B = \pi_k$, construct the Lyapunov function $L_{\epsilon_{k+1}}$, where $\epsilon_{k+1}$ is a solution to (5);
**end for**
**Return** Final policy $\pi_{k^*}$

---

## 4.1 Lyapunov-based Safe RL Algorithms

In order to improve scalability of SVI and SPI, we develop two *off-policy* safe RL algorithms, namely safe DQN and safe DPI, which replace the value and policy updates in safe DP with function approximations. Their pseudo-codes can be found in Appendix D. Before going into their details, we first introduce the policy distillation method, which will be later used in the safe RL algorithms.

**Policy Distillation:** Consider the following LP problem for policy optimization in SVI and SPI:

$$\pi'(\cdot|x) \in \arg\min_{\pi \in \Delta}\left\{\pi(\cdot|x)^\top Q(x, \cdot) : (\pi(\cdot|x) - \pi_B(\cdot|x))^\top Q_L(x, \cdot) \leq \widetilde{\epsilon}'(x)\right\}, \qquad (6)$$

where $Q_L(x, a) = d(x) + \widetilde{\epsilon}'(x) + \sum_{x'} P(x'|x, a)L_{\widetilde{\epsilon}'}(x')$ is the state-action Lyapunov function. When the state-space is large (or continuous), we shall use function approximation. Consider a parameterized policy $\pi_\phi$ with weights $\phi$. Utilizing the distillation concept [36], after computing the optimal action probabilities w.r.t. a batch of states, the policy $\pi_\phi$ is updated by solving $\phi^* \in \arg\min_\phi \frac{1}{m}\sum_{m=1}^{M}\sum_{t=0}^{\overline{\mathrm{T}}-1} D_{\mathrm{JSD}}(\pi_\phi(\cdot|x_{t,m}) \| \pi'(\cdot|x_{t,m}))$, where $D_{\mathrm{JSD}}$ is the Jensen-Shannon divergence. Pseudo-code of distillation is given in Algorithm 3 in Appendix D.

**Safe $Q-$learning (SDQN):** Here we sample an off-policy mini-batch of state, action, cost, and next-state from the replay buffer, and use it to update the value function estimates that minimize the MSE losses of the Bellman residuals. We first construct the state-action Lyapunov function estimate $\widehat{Q}_L(x, a; \theta_D, \theta_T) = \widehat{Q}_D(x, a; \theta_D) + \widetilde{\epsilon}' \cdot \widehat{Q}_T(x, a; \theta_T)$ by learning the constraint value network $\widehat{Q}_D$ and stopping time value network $\widehat{Q}_T$. With a current baseline policy $\pi_k$, one can use function approximation to approximate the auxiliary constraint cost (which is the solution to (5)) by $\widetilde{\epsilon}'(x) = \widetilde{\epsilon}' = (d_0 - \pi_k(\cdot|x_0)^\top \widehat{Q}_D(x_0, \cdot; \theta_D))/\pi_k(\cdot|x_0)^\top \widehat{Q}_T(x_0, \cdot; \theta_T)$. Equipped with the Lyapunov function, at each iteration, one can do a standard DQN update, except that the optimal action probabilities are computed by solving (6). Details of SDQN is given in Algorithm 4 in Appendix D.

**Safe Policy Improvement (SDPI):** Similar to SDQN, in this algorithm, we first sample an off-policy mini-batch from the replay buffer and use it to update the value function estimates (w.r.t. objective, constraint, and stopping-time estimate) that minimize MSE losses. Different from SDQN, in SDPI the value estimation is done using policy evaluation, which means that the objective $Q-$function is trained to minimize the Bellman residual w.r.t. actions generated by the current policy $\pi_k$, instead of the greedy actions. Using the same construction as in SDQN for auxiliary cost $\widetilde{\epsilon}'$ and state-action Lyapunov function $\widehat{Q}_L$, we then perform a policy improvement step by computing

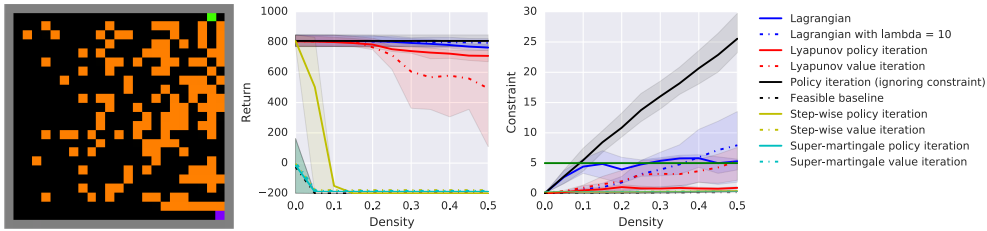

Figure 1: Results of various planning algorithms on the grid-world environment with obstacles, with x-axis showing the obstacle density. From the leftmost column, the first figure illustrates the 2D planning domain example ($\rho = 0.25$). The second and the third figures show the average return and the average cumulative constraint cost of the CMDP methods, respectively. The fourth figure displays all the methods used in the experiment. The shaded regions indicate the $80\%$ confidence intervals. Clearly the safe DP algorithms compute policies that are safe and have good performance.

a set of greedy action probabilities from (6) and constructing an updated policy $\pi_{k+1}$ using policy distillation. Assuming both value and policy approximations have low error, SDPI resembles several interesting properties of SPI, such as maintaining safety during training and monotonically improving the policy. To improve learning stability, instead of the full policy update, one can further consider a partial update $\pi_{k+1} = (1-\alpha)\pi_k + \alpha\pi'$, where $\alpha \in (0,1)$ is a *mixing constant* that controls safety and exploration [2, 19]. Details of SDPI is summarized in Algorithm 5 in Appendix D.

## 5 Experiments

Motivated by the safety issues of RL in [22], we validate our safe RL algorithms using a stochastic 2D grid-world motion planning problem. In this domain, an agent (e.g., a robotic vehicle) starts in a safe region and its objective is to travel to a given destination. At each time step, the agent can move to any of its four neighboring states. Due to sensing and control noise, however, with probability $\delta$ a move to a random neighboring state occurs. To account for fuel usage, the stage-wise cost of each move until reaching the destination is 1, while the reward achieved for reaching the destination is 1000. Thus, we would like the agent to reach the destination in the shortest possible number of moves. In between the starting and destination points, there are number of obstacles that the agent may pass through but should avoid for safety; each time the agent hits an obstacle it incurs a constraint cost of 1. Thus, in the CMDP setting, the agent's goal is to reach the destination in the shortest possible number of moves, while hitting the obstacles at most $d_0$ times or less. For demonstration purposes, we choose a $25 \times 25$ grid-world (see Figure 1) with a total of 625 states. We also have a density ratio $\rho \in (0,1)$ that sets the obstacle-to-terrain ratio. When $\rho$ is close to 0, the problem is obstacle-free, and if $\rho$ is close to 1, then the problem becomes more challenging. In the normal problem setting, we choose a density $\rho = 0.3$, an error probability $\delta = 0.05$, a constraint threshold $d_0 = 5$, and a maximum horizon of 200 steps. The initial state is located in $(24, 24)$ and the goal is placed in $(0, \alpha)$, where $\alpha \in [0, 24]$ is a uniform random variable. To account for statistical significance, the results of each experiment are averaged over 20 trials.

**CMDP Planning:** In this task, we have explicit knowledge of the reward function and transition probability. The main goal is to compare our safe DP algorithms (SPI and SVI) with the following common CMDP baseline methods: (i) *Step-wise Surrogate*, (ii) *Super-martingale Surrogate*, (iii) *Lagrangian*, and (iv) *Dual LP*. Since the methods in (i) and (ii) are surrogate algorithms, we will also evaluate these methods with both value iteration and policy iteration. To illustrate the level of sub-optimality, we will also compare the returns and constraint costs of these methods with baselines that are generated by maximizing return or minimizing constraint cost of two separate MDPs. The main objective here is to illustrate that safe DP algorithms are less conservative than other surrogate methods, are more numerically stable than the Lagrangian method, and are more computationally efficient than the Dual LP method (see Appendix F), without using function approximations.

Figure 1 presents the results on returns and cumulative constraint costs of the aforementioned CMDP methods over a spectrum of $\rho$ values, ranging from 0 to 0.5. In each method, the initial policy is a conservative baseline policy $\pi_B$ that minimizes the constraint cost. The empirical results indicate that although the polices generated by the four surrogate algorithms are feasible, they do not have significant policy improvement, i.e., return values are close to that of the initial baseline policy. Over all density settings, the SPI algorithm consistently computes a solution that is feasible and has good performance. The policy returned by SVI is always feasible and has near-optimal performance

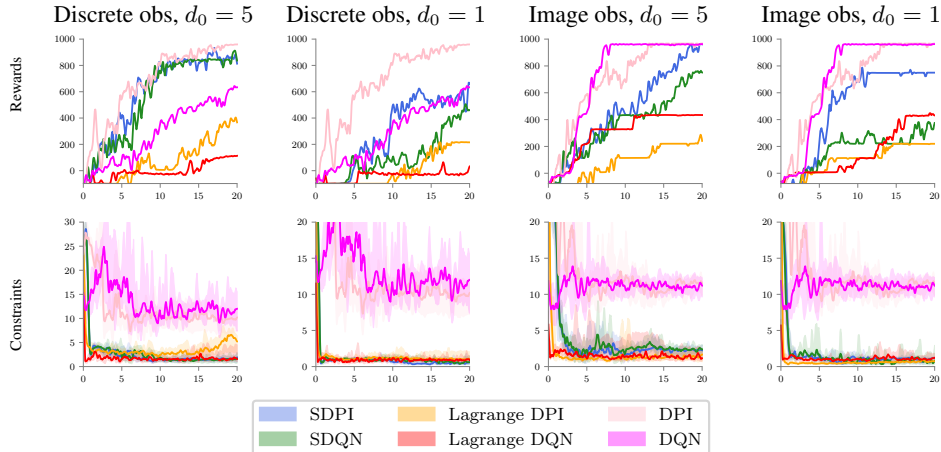

Figure 2: Results of various RL algorithms on the grid-world environment with obstacles, with x-axis in thousands of episodes. We include runs using discrete observations (a one-hot encoding of the agent's position) and image observations (showing the entire RGB 2D map of the world). We discover that the Lyapunov-based approaches can perform safe learning, despite the fact that the model of the environment is not known and that deep function approximation is necessary.

when the obstacle density is low. However, due to numerical instability, its performance degrades as $\rho$ grows. Similarly, the Lagrangian methods return a near-optimal solution over most settings, but due to numerical issues their solutions start to violate constraint as $\rho$ grows.

**Safe Reinforcement Learning:** Here we present the results of RL algorithms on this safety task. We evaluate their learning performance on two variants: one in which the observation is a one-hot encoding of the agent's location, and the other in which the observation is the 2D image representation of the grid map. In each of these, we evaluate performance when $d_0 = 1$ and $d_0 = 5$. We compare our proposed safe RL algorithms, SDPI and SDQN, with their unconstrained counterparts, DPI and DQN, as well as the Lagrangian approach to safe RL, in which the Lagrange multiplier is optimized via extensive grid search. Details of the experimental setup are given in Appendix F. To make the tasks more challenging, we initialize the RL algorithms with a randomized baseline policy.

Figure 2 shows the results of these methods across all task variants. We observe that SDPI and SDQN can adequately solve the tasks and compute good return performance (similar to that of DQN and DPI in some cases), while guaranteeing safety. Another interesting observation in the SDQN and SDPI algorithms is that, once the algorithm finds a safe policy, then all updated policies *remain* safe throughout training. On the contrary, the Lagrangian approach often achieves worse rewards and is more apt to violate the constraints during training, [6], and the performance is very sensitive to the initial conditions. Furthermore, in some cases (in experiment with $d_0 = 5$ and with discrete observations) the Lagrangian method cannot guarantee safety throughout training.

# 6 Conclusion

In this paper, we formulated the problem of safe RL as a CMDP and proposed a *novel* Lyapunov approach to solve CMDPs. We also derived an effective LP-based method to generate Lyapunov functions, such that the corresponding algorithm guarantees feasibility and optimality under certain conditions. Leveraging these theoretical underpinnings, we showed how Lyapunov approaches can be used to transform DP (and RL) algorithms into their safe counterparts that only require straightforward modifications in the algorithm implementations. We empirically validated our theoretical findings in using the Lyapunov approach to guarantee safety and robust learning in RL. In general, our work represents a step forward in deploying RL to real-world problems in which guaranteeing safety is of paramount importance. Future research will focus on two directions. On the algorithmic perspective, one major extension is to apply the Lyapunov approach to policy gradient algorithms and compare its performance with CPO in continuous action problems. On the practical aspect, future work includes evaluating the Lyapunov-based RL algorithms on several real-world testbeds.

## Footnotes

[1]One example of $\pi_B$ is a policy that minimizes the constraint, i.e., $\pi_B(\cdot|x) \in \arg\min_{\pi \in \Delta(x)} \mathcal{D}_\pi(x)$.

[2]To see this, the constraint cost function $\mathcal{D}_{\pi_B}(x)$ is a valid Lyapunov function, i.e., $\mathcal{D}_{\pi_B}(x_0) \leq d_0$, $\mathcal{D}_{\pi_B}(x) = 0, \forall x \in \mathcal{X} \backslash \mathcal{X}'$, and $\mathcal{D}_{\pi_B}(x) = T_{\pi_B, d}[\mathcal{D}_{\pi_B}](x) = \mathbb{E}\left[\sum_{t=0}^{\mathrm{T}^*-1} d(x_t) \mid \pi_B, x\right], \ \forall x \in \mathcal{X}'$.

[3]The definition of total variation distance is given by $D_{TV}(\pi^*||\pi_B)(x) = \frac{1}{2}\sum_{a \in \mathcal{A}} |\pi_B(a|x) - \pi^*(a|x)|$.

[4]This is due to the fact that $d_0 - \mathcal{D}_{\pi_B}(x_0) \geq 0$, and thus, $\widetilde{\epsilon}(x) = 0$ is a feasible solution.

[5]The strict concavity property in the objective function is mainly for the purpose of tie-breaking. One standard example is the entropy regularizer with a small regularization term.

[6]In Appendix F, we also report the results from the Lagrangian method in which the Lagrange multiplier is learned using gradient ascent method [10] and we observe similar (or even worse) behaviors.

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
