[Supplementary Material · safety-pages-11-28 (1).pdf]

# A Safety Constraints in Planning Problems

To motivate the CMDP formulation studied in this paper, in this section we include two real-life examples of modeling safety using the reachability constraint, and the constraint that limits the agent's visits to undesirable states.

## A.1 Reachability Constraint

Reachability is a common concept in motion-planning and engineering applications, where for any given policy $\pi$ and initial state $x_0$, the following the constraint function is considered:

$$\mathbb{P}(\exists t \in \{0, 1, \ldots, \mathrm{T}^* - 1\}, x_t \in \mathcal{S}_H \mid x_0, \pi).$$

Here $\mathcal{S}_H$ represents the real subset of hazardous regions for the states and actions. Therefore, the constraint cost represents the probability of reaching an unsafe region at any time before the state reaches the terminal state. To further analyze this constraint function, one notices that

$$\mathbb{P}(\exists t \in \{0, 1, \ldots, \mathrm{T}^* - 1\}, x_t \in \mathcal{S}_H \mid x_0, \pi) = \mathbb{P}\left(\bigcup_{t=0}^{\mathrm{T}^*-1} \bigcap_{j=0}^{t-1} \{x_j \notin \mathcal{S}_H\} \cap \{x_t \in \mathcal{S}_H\} \mid x_0, \pi\right)$$

$$= \mathbb{E}\left[\sum_{t=0}^{\mathrm{T}^*-1} \prod_{j=0}^{t-1} \mathbf{1}\{x_j \notin \mathcal{S}_H\} \cdot \mathbf{1}\{x_t \in \mathcal{S}_H\} \mid x_0, \pi\right].$$

In this case, a policy $\pi$ is deemed *safe* if the reachability probability to the unsafe region is bounded by threshold $d_0 \in (0, 1)$, i.e.,

$$\mathbb{P}(\exists t \in \{0, 1, \ldots, \mathrm{T}^* - 1\}, x_t \in \mathcal{S}_H \mid x_0, \pi) \leq d_0. \tag{7}$$

To transform the reachability constraint into a standard CMDP constraint, we define an additional state $s \in \{0, 1\}$ that keeps track of the reachability status at time $t$. Here $s_t = 1$ indicates the system has never visited a hazardous region up till time $t$, and otherwise $s_t = 0$. Let $s_0 = 1$, we can easily see that by defining the following deterministic transition

$$s_t = s_{t-1} \cdot \mathbf{1}\{x_t \notin \mathcal{S}_H\}, \ \forall t \geq 1,$$

$s_t$ has the following formulation: $s_t = \prod_{j=0}^{t-1} \mathbf{1}\{x_j \notin \mathcal{S}_H\}$.

Collectively, with the state augmentation $\hat{x} = (x, s)$, one defines the augmented CMDP $(\hat{\mathcal{X}}, \mathcal{A}, C, \hat{D}, \hat{P}, \hat{x}_0, d_0)$, where $\hat{\mathcal{X}} = \mathcal{X} \times \{0, 1\}$ is the augmented state space, $\hat{d}(\hat{x}) = s \cdot d(x)$ is the augmented constraint cost, $\hat{P}(\hat{x}'|\hat{x}, a) = P(x'|x, a) \cdot \mathbf{1}\{s' = s \cdot \mathbf{1}\{x \notin \mathcal{S}_H\}\}$ is the augmented transition probability, and $\hat{x}_0 = (x_0, 1)$ is the initial (augmented) state. By using this augmented CMDP, immediately the reachability constraint is equivalent to $\mathbb{E}\left[\sum_{t=0}^{\mathrm{T}^*-1} \hat{d}(\hat{x}_t) \mid x_0, \pi\right] \leq d_0$.

## A.2 Constraint w.r.t. Undesirable Regions of States

Consider the notion of safety where one restricts the total visiting frequency of an agent to an undesirable region (of states). This notion of safety appears in applications such as system maintenance, in which the system can only tolerate its state to visit (in expectation) a hazardous region, namely $\mathcal{S}_H$, for a fixed number of times. Specifically, for given initial state $x_0$, consider the following constraint that bounds the total frequency of visiting $\mathcal{S}_H$ with a pre-defined threshold $d_0$, i.e., $\mathbb{E}\left[\sum_{t=0}^{\mathrm{T}^*-1} d(x_t) \mid x_0, \pi\right] \leq d_0$, where $d(x_t) = \mathbf{1}\{x_t \in \mathcal{S}_H\}$. To model this notion of safety using a CMDP, one can rewrite the above constraint using the constraint immediate cost $d(x) = \mathbf{1}\{x \in \mathcal{S}_H\}$, and the constraint threshold $d_0$. To study the connection between the reachability constraint, and the above constraint w.r.t. undesirable region, notice that

$$\mathbb{E}\left[\sum_{t=0}^{\mathrm{T}^*-1} \prod_{j=0}^{t-1} \mathbf{1}\{x_j \notin \mathcal{S}_H\} \cdot \mathbf{1}\{x_t \in \mathcal{S}_H\} \mid x_0, \pi\right] \leq \mathbb{E}\left[\sum_{t=0}^{\mathrm{T}^*-1} \mathbf{1}\{x_t \in \mathcal{S}_H\} \mid x_0, \pi\right].$$

This clearly indicates that any policies which satisfies the constraint w.r.t. undesirable region, also satisfies the reachability constraint.

# B Existing Approaches for Solving CMDPs

Before going to the main result, we first revisit several existing CMDP algorithms in the literature, which later serve as the baselines for comparing with our safe CMDP algorithms. For the sake of brevity, we will only provide an overview of these approaches here and defer their details to Appendix B.1.

**The Lagrangian Based Algorithm:** The standard way of solving problem $\mathcal{OPT}$ is by applying the Lagrangian method. To start with, consider the following minimax problem: $\min_{\pi \in \Delta} \max_{\lambda \geq 0} \ \mathcal{C}_\pi(x_0) + \lambda(\mathcal{D}_\pi(x_0) - d_0)$, where $\lambda$ is the Lagrange multiplier w.r.t. the CMDP constraint. According to Theorem 9.9 and Theorem 9.10 in [3], the optimal policy $\pi^*$ of problem $\mathcal{OPT}$ can be calculated by solving the following Lagrangian function $\pi^* \in \arg\min_{\pi \in \Delta} \mathcal{C}_\pi(x_0) + \lambda^*_{d_0} \mathcal{D}_\pi(x_0)$, where $\lambda^*_{d_0}$ is the optimal Lagrange multiplier. Utilizing this result, one can compute the saddle point pair $(\pi^*, \lambda^*)$ using primal-dual iteration. Specifically, for a given $\lambda \geq 0$, solve the policy minimization problem using standard dynamic programming with $\lambda-$parametrized Bellman operator $T_\lambda[V](x) = \min_{\pi \in \Delta(x)} T_{\pi, c+\lambda d}[V](x)$ if $x \in \mathcal{X}$; For a given policy $\pi$, solve for the following linear optimization problem: $\max_{\lambda \geq 0} \ \mathcal{C}_\pi(x_0) + \lambda(\mathcal{D}_\pi(x_0) - d_0)$. Based on Theorem 9.10 in [3], this procedure will asymptotically converge to the saddle point solution. However, this algorithm presents several major challenges. (i) In general there is no known convergence rate guarantees, several studies [21] also showed that using primal-dual first-order iterative method to find saddle point may run into numerical instability issues; (ii) Choosing a good initial estimate of the Lagrange multiplier is not intuitive; (iii) Following the same arguments from [2], during iteration the policy may be infeasible w.r.t. problem $\mathcal{OPT}$, and feasibility is guaranteed after the algorithm converges. This is hazardous in RL when one needs to execute the intermediate policy (which may be unsafe) during training.

**The Dual LP Based Algorithm:** Another method of solving problem $\mathcal{OPT}$ is based on computing its *occupation measures* w.r.t. the optimal policy. In transient MDPs, for any given policy $\pi$ and initial state $x_0$, the state-action occupation measure is $\rho_\pi(x, a) = \mathbb{E}\left[\sum_{t=0}^{T^*-1} \mathbf{1}\{x_t = x, a_t = a\} \mid x_0, \pi\right]$, which characterizes the total visiting probability of state-action pair $(x, a) \in \mathcal{X} \times \mathcal{A}$, induced by policy $\pi$ and initial state $x_0$. Utilizing this quantity, Theorem 9.13 in [3], has shown that problem $\mathcal{OPT}$ can be reformulated as a linear programming (LP) problem (see Equation (8) to (9) in Appendix B.1), whose decision variable is of dimension $|\mathcal{X}'||\mathcal{A}|$, and it has $2|\mathcal{X}'||\mathcal{A}|+1$ constraints. Let $\rho^*$ be the solution of this LP, the optimal Markov stationary policy is given by $\pi^*(a|x) = \rho^*(x, a)/\sum_{a \in \mathcal{A}} \rho^*(x, a)$. To solve this problem, one can apply the standard algorithm such as interior point method, which is a strong polynomial time algorithm with complexity $O(|\mathcal{X}'|^2|\mathcal{A}|^2(2|\mathcal{X}'||\mathcal{A}| + 1))$ [8]. While this is a straight-forward methodology, it can only handle CMDPs with finite state and action spaces. Furthermore, this approach is computationally expensive when the size of these spaces are large. To the best of our knowledge, it is also unclear how to extend this approach to RL, when transition probability and immediate reward/constraint reward functions are unknown.

**Step-wise Constraint Surrogate Approach:** This approach transforms the multi-stage CMDP constraint into a sequence of step-wise constraints, where each step-wise constraint can be directly embedded into set of admissible actions in the Bellman operator. To start with, for any state $x \in \mathcal{X}'$, consider the following feasible set of policies: $\mathcal{F}^{\text{SW}}(d_0, x) = \{\pi \in \Delta : \sum_{a \in \mathcal{A}} \sum_{x' \in \mathcal{X}'} \pi(a|x)P(x'|x, a)d(x') \leq \frac{d_0}{\overline{T}}\}$, where $\overline{T}$ is the upper-bound of the MDP stopping time. Based on (10) in Appendix B.1, one deduces that every policy $\pi$ in $\bigcap_{x \in \mathcal{X}'} \mathcal{F}^{\text{SW}}(d_0, x)$ is a feasible policy w.r.t. problem $\mathcal{OPT}$. Motivated by this observation, a solution policy can be solved by $\min_{\pi \in \bigcap_{x \in \mathcal{X}'} \mathcal{F}^{\text{SW}}(d_0, x)} \mathbb{E}\left[\sum_{t=0}^{T^*-1} c(x_t, a_t) \mid x_0, \pi\right]$. One benefit of studying this surrogate problem is that its solution satisfies the Bellman optimality condition w.r.t. the step-wise Bellman operator as $T^{\text{SW}}[V](x) = \min_{\pi \in \mathcal{F}^{\text{SW}}(d_0, x)} \sum_a \pi(a|x)\left[c(x, a) + \sum_{x' \in \mathcal{X}'} P(x'|x, a)V(x')\right]$ for any $x \in \mathcal{X}'$. In particular $T^{\text{SW}}$ is a contraction operator, which implies that there exists a unique solution $V^{*, \text{SW}}$ to fixed point equation $T^{\text{SW}}[V](x) = V(x)$ for $x \in \mathcal{X}'$ such that $V^{*, \text{SW}}(x_0)$ is a solution to the surrogate problem. Therefore this problem can be solved by standard DP methods such as value iteration or policy iteration. Furthermore, based on the structure of $\mathcal{F}^{\text{SW}}(d_0, x)$, any surrogate policy

is feasible w.r.t. problem $\mathcal{OPT}$. However, the major drawback is that the step-wise constraint in $\mathcal{F}^{\mathrm{SW}}(d_0, x)$ can be much more stringent than the original safety constraint in problem $\mathcal{OPT}$.

**Super-martingale Constraint Surrogate Approach:** This surrogate algorithm is originally proposed by [14], where the CMDP constraint is reformulated as the surrogate value function $\mathcal{DS}_\pi(x) = \max\{d_0, \mathcal{D}_\pi(x)\}$ at initial state $x \in \mathcal{X}'$. It has been shown that an arbitrary policy $\pi$ is a feasible policy of the CMDP if and only if $\mathcal{DS}_\pi(x_0) = d_0$. Notice that $\mathcal{DS}_\pi$ is known as a *super-martingale surrogate*, due to the inequality $\mathcal{DS}_\pi(x) \leq T_\pi^{DS}[\mathcal{DS}_\pi](x)$ with respect to the contraction Bellman operator $T_\pi^{DS}[V](x) = \sum_{a \in \mathcal{A}} \pi(a|x) \max\left\{d_0, d(x) + \sum_{x' \in \mathcal{X}'} P(x'|x,a)V(x')\right\}$ of the constraint value function. However, for arbitrary policy $\pi$, in general it is non-trivial to compute the value function $\mathcal{DS}_\pi(x)$, and instead one can easily compute its upper-bound value function $\overline{\mathcal{DS}}_\pi(x)$ which is the solution of the fixed-point equation $V(x) = T_\pi^{DS}[V](x)$, $\forall x \in \mathcal{X}'$, using standard dynamic programming techniques. To better understand how this surrogate value function guarantees feasibility in problem $\mathcal{OPT}$, at each state $x \in \mathcal{X}'$ consider the optimal value function of the minimization problem $\overline{\mathcal{DS}}(x) = \min_{\pi \in \Delta} \overline{\mathcal{DS}}_\pi(x)$. Then whenever $\overline{\mathcal{DS}}(x_0) \leq d_0$, the corresponding solution policy $\pi$ is a feasible policy of problem $\mathcal{OPT}$, i.e., $\mathcal{D}_\pi(x_0) \leq \overline{\mathcal{DS}}_\pi(x_0) = d_0$. Now define $\mathcal{F}^{\mathrm{DS}}(x) := \{\pi \in \Delta : T_\pi^{DS}[\overline{\mathcal{DS}}](x) = \overline{\mathcal{DS}}(x)\}$ as the set of refined feasible policies induced by $\overline{\mathcal{DS}}$. If the condition $\overline{\mathcal{DS}}(x_0) \leq d_0$ holds, then all the policies in $\mathcal{F}^{\mathrm{DS}}(x)$ are feasible w.r.t. problem $\mathcal{OPT}$. Utilizing this observation, a surrogate solution policy of problem $\mathcal{OPT}$ can be found by computing the solution policy of the fixed-point equation $T_{\mathcal{F}^{\mathrm{DS}}}[V](x) = V(x)$, for $x \in \mathcal{X}'$, where $T_{\mathcal{F}^{\mathrm{DS}}}[V](x) = \min_{\pi \in \mathcal{F}^{\mathrm{DS}}(d_0, x)} T_{\pi,c}[V](x)$. Notice that $T_{\mathcal{F}^{\mathrm{DS}}}$ is a contraction operator, this procedure can also be solved using standard DP methods. The major benefit of this 2-step approach is that the computation of the feasibility set is decoupled from solving the optimization problem. This allows us to apply approaches such as the *lexicographical ordering* method from multi-objective stochastic optimal control methods [35] to solve the CMDP, for which the constraint value function has a higher lexicographical order than the objective value function. However, since the refined set of feasible policies is constructed prior to policy optimization, it might still be overly conservative. Furthermore, even if there exists a non-trivial solution policy to the surrogate problem, characterizing its sub-optimality performance bound remains a challenging task.

## B.1 Details of Existing Solution Algorithms

In this section, we provide the details of the existing algorithms for solving CMDPs.

**The Lagrangian Based Algorithm:** The standard way of solving problem $\mathcal{OPT}$ is by applying the Lagrangian method. To start with, consider the following minimax problem:

$$\min_{\pi \in \Delta} \max_{\lambda \geq 0} \; \mathcal{C}_\pi(x_0) + \lambda(\mathcal{D}_\pi(x_0) - d_0),$$

where $\lambda$ is the Lagrange multiplier of the CMDP constraint, and the Lagrangian function is given by

$$\mathcal{L}_{x_0, d_0}(\pi, \lambda) = \mathcal{C}_\pi(x_0) + \lambda(\mathcal{D}_\pi(x_0) - d_0) = \mathbb{E}\left[\sum_{t=0}^{\mathrm{T}^*-1} c(x_t, a_t) + \lambda d(x_t) \mid \pi, x_0\right] - \lambda d_0.$$

A solution pair $(\pi^*, \lambda^*)$ is considered as a saddle point of Lagrangian function $\mathcal{L}_{x_0, d_0}(\pi, \lambda)$ if the following condition holds:

$$\mathcal{L}_{x_0, d_0}(\pi^*, \lambda) \leq L_{x_0, d_0}(\pi, \lambda) \leq \mathcal{L}_{x_0, d_0}(\pi, \lambda^*), \; \forall \pi, \lambda \geq 0.$$

According to Theorem 9.10 in [3], suppose the interior set of feasible set of problem $\mathcal{OPT}$ is non-empty, then there exists a solution pair $(\pi^*, \lambda^*)$ to the minimax problem that is a saddle point of Lagrangian function $\mathcal{L}_{x_0, d_0}(\pi, \lambda)$. Furthermore, Theorem 9.9 in [3] shows that strong duality holds:

$$\min_{\pi \in \Delta} \max_{\lambda \geq 0} \; \mathcal{L}_{x_0, d_0}(\pi, \lambda) = \max_{\lambda \geq 0} \min_{\pi \in \Delta} \; \mathcal{L}_{x_0, d_0}(\pi, \lambda).$$

This implies that the optimal policy $\pi^* \in \Delta$ can be calculated by solving the following Lagrangian function $\pi^* \in \arg\min_{\pi \in \Delta} \mathcal{L}_{x_0, d_0}(\pi, \lambda^*)$, with optimal Lagrange multiplier $\lambda^*$.

Utilizing the structure of the Lagrangian function $\mathcal{L}_{x_0,d_0}(\pi, \lambda)$, for any fixed Lagrange multiplier $\lambda \geq 0$, consider the $\lambda-$Bellman operator $T_\lambda[V]$, where

$$T_\lambda[V](x) = \begin{cases} \min_{\pi \in \Delta(x)} T_{\pi,c+\lambda d}[V](x) & \text{if } x \in \mathcal{X}', \\ 0 & \text{otherwise.} \end{cases}$$

Since $T_\lambda[V]$ is a $\gamma-$contraction operator, by the Bellman principle of optimality, there is a unique solution $V^*$ to the fixed point equation $T_\lambda[V](x) = V(x)$ for $x \in \mathcal{X}$, which can be solved by dynamic programming algorithms, such as value iteration or policy iteration. Furthermore, the $\lambda-$optimal policy $\pi_\lambda^*$ has the form of

$$\pi_\lambda^*(\cdot|x) \in \arg \min_{\pi \in \Delta(x)} T_{\pi,c+\lambda d}[V^*](x) \quad \text{if } x \in \mathcal{X}',$$

and $\pi_\lambda^*(\cdot|x)$ is an arbitrary probability distribution function if $x \notin \mathcal{X}'$.

**The Dual LP Based Algorithm:**  The other commonly-used method for solving problem $\mathcal{OPT}$ is based on computing its *occupation measures* w.r.t. the optimal policy. In a transient MDP, for any given policy $\pi$ and initial state $x_0 \in \mathcal{X}'$ the state-action occupation measure is defined as

$$\rho_{\pi,x_0}(x,a) = \mathbb{E}\left[ \sum_{t=0}^{T^*-1} \mathbf{1}\{x_t = x, a_t = a\} \mid x_0, \pi \right], \quad \forall x \in \mathcal{X}, \ \forall a \in \mathcal{A},$$

and the occupation measure at state $x \in \mathcal{X}$ is defined as $\rho_{\pi,x_0}(x) = \sum_{a \in \mathcal{A}} \rho_{\pi,x_0}(x,a)$. Clearly these two occupation measures are related by the following property: $\rho_{\pi,x_0}(x,a) = \rho_{\pi,x_0}(x) \cdot \pi(a|x)$. Furthermore, using the fact that a occupation measure $\rho_{\pi,x_0}(x,a)$ is indeed the sum of visiting distribution of the transient MDP induced by policy $\pi$, one clearly sees that it satisfies the following set of constraints:

$$\mathcal{Q}(x_0) = \left\{ \rho : \mathcal{X}' \times \mathcal{A} \to \mathbb{R} : \rho(x,a) \geq 0, \forall x \in \mathcal{X}', a \in \mathcal{A}, \right.$$

$$\left. \sum_{x_p \in \mathcal{X}', a \in \mathcal{A}} \rho(x_p, a)(\mathbf{1}\{x_p = x\} - P(x|x_p, a)) = \mathbf{1}\{x = x_0\} \right\}.$$

Therefore, by Theorem 9.13 in [3], equivalently problem $\mathcal{OPT}$ can be solved by the LP optimization problem with $2|\mathcal{X}'||\mathcal{A}| + 1$ constraints:

$$\min_{\rho \in \mathcal{Q}(x_0)} \sum_{x \in \mathcal{X}', a \in \mathcal{A}} \rho(x,a) c(x,a) \tag{8}$$

$$\text{subject to} \sum_{x \in \mathcal{X}', a \in \mathcal{A}} \rho(x,a) d(x) \leq d_0, \tag{9}$$

and equipped with the minimizer minimizer $\rho^* \in \mathcal{Q}(x_0)$, the (non-uniform) optimal Markovian stationary policy is given by the following form:

$$\pi^*(a|x) = \frac{\rho^*(x,a)}{\sum_{a \in \mathcal{A}} \rho^*(x,a)}, \quad \forall x \in \mathcal{X}', \ \forall a \in \mathcal{A}.$$

**Step-wise Constraint Surrogate Approach:**  To start with, without loss of generality assume that the agent is safe at the initial phase, i.e., $d(x_0) \leq 0$. For any state $x \in \mathcal{X}'$, consider the following feasible set of policies:

$$\mathcal{F}^{\text{SW}}(d_0, x) = \left\{ \pi \in \Delta(x) : \sum_{a \in \mathcal{A}} \sum_{x' \in \mathcal{X}'} \pi(a|x) P(x'|x,a) d(x') \leq \frac{d_0}{\overline{T}} \right\},$$

where $\overline{T}$ is the uniform upper-bound of the random stopping time in the transient MDP. Immediately, for any policy $\pi \in \bigcap_{x \in \mathcal{X}'} \mathcal{F}^{\text{SW}}(d_0, x)$, one has the following inequality:

$$\mathbb{E}\left[ \sum_{t=0}^{T^*-1} d(x_t) \mid \pi, x_0 \right] = d(x_0) + \sum_{x \in \mathcal{X}', a \in \mathcal{A}} \rho_{\pi,x_0}(x,a) \sum_{x' \in \mathcal{X}'} P(x'|x,a) d(x')$$

$$\leq \mathbb{E}\left[ \sum_{t=0}^{T^*-1} \frac{d_0}{\overline{T}} \mid \pi, x_0 \right] \leq d_0, \tag{10}$$

where $\rho_{\pi,x_0}(x,a) = \mathbb{E}\left[\sum_{t=0}^{\mathrm{T}^*-1} \mathbf{1}\{x_t = x\}|\pi, x_0\right]$ is the state-action occupation measure with initial state $x_0$ and policy $\pi$, which implies that the policy $\pi$ is safe, i.e., $\mathcal{D}_\pi(x_0) \le d_0$. Equipped with this property, we propose the following surrogate problem for problem $\mathcal{OPT}$, whose solution (if exists) is guaranteed to be safe:

> **Problem $\mathcal{OPT}^{\mathrm{SW}}$:** Given an initial state $x_0$, a threshold $d_0$, solve $\min_{\pi \in \bigcap_{x \in \mathcal{X}'} \mathcal{F}^{\mathrm{SW}}(d_0,x)} \mathbb{E}\left[\sum_{t=0}^{\mathrm{T}^*-1} c(x_t, a_t) \mid x_0, \pi\right].$

To solve problem $\mathcal{OPT}^{\mathrm{SW}}$, for each state $x \in \mathcal{X}'$ define the step-wise Bellman operator as

$$T^{\mathrm{SW}}[V](x) = \min_{\pi \in \mathcal{F}^{\mathrm{SW}}(d_0,x)} T_{\pi,c}[V](x).$$

Based on the standard arguments in [7], the Bellman operator $T^{\mathrm{SW}}$ is a contraction operator, which implies that there exists a unique solution $V^{*,\mathrm{SW}}$ to the fixed-point equation $T^{\mathrm{SW}}[V](x) = V(x)$, for any $x \in \mathcal{X}'$, such that $V^{*,\mathrm{SW}}(x_0)$ is a solution to problem $\mathcal{OPT}^{\mathrm{SW}}$.

**Super-martingale Constraint Surrogate Approach:** The following surrogate algorithm is proposed by [14]. Before going to the main algorithm, first consider the following surrogate constraint value function w.r.t. policy $\pi \in \Delta$ and state $x \in \mathcal{X}'$:

$$\mathcal{DS}_\pi(x) = \max\left\{d_0, \mathcal{D}_\pi(x)\right\} = \max\left\{d_0, d(x) + \sum_{a \in \mathcal{A}} \pi(a|x) \sum_{x' \in \mathcal{X}'} P(x'|x,a)\mathcal{D}_\pi(x')\right\}$$

$$\le \sum_{a \in \mathcal{A}} \pi(a|x)\max\left\{d_0, d(x) + \sum_{x' \in \mathcal{X}'} P(x'|x,a)\mathcal{D}_\pi(x')\right\}.$$

The last inequality is due to the fact that the $\max$ operator is convex. Clearly, by definition one has

$$\mathcal{DS}_\pi(x) \ge \mathcal{D}_\pi(x) \text{ for each } x \in \mathcal{X}'.$$

On the other hand, one also has the following property: $\mathcal{DS}_\pi(x_0) = d_0$ if and only if the constraint of problem $\mathcal{OPT}$ is satisfied, i.e., $\mathcal{D}_\pi(x_0) \le d_0$. Now, by utilizing the contraction operator

$$T_\pi^{DS}[V](x) = \sum_{a \in \mathcal{A}} \pi(a|x)\max\left\{d_0, d(x) + \sum_{x' \in \mathcal{X}'} P(x'|x,a)V(x')\right\},$$

w.r.t. policy $\pi$, and by utilizing the definition of the constraint value function $\mathcal{DS}_\pi$, one immediately has the chain of inequalities:

$$\mathcal{DS}_\pi(x) \le T_\pi^{DS}[\mathcal{D}_\pi](x) \le T_\pi^{DS}[\mathcal{DS}_\pi](x),$$

which implies that the value function $\{\mathcal{DS}_\pi(x)\}_{x \in \mathcal{X}'}$ is a *super-martingale*, i.e., $\mathcal{DS}_\pi(x) \le T_\pi^{DS}[\mathcal{DS}_\pi](x)$, for $x \in \mathcal{X}'$.

However in general the constraint value function of interest, i.e., $\mathcal{DS}_\pi$, cannot be directly obtained as the solution of fixed point equation. Thus in what follows, we will work with its approximation $\overline{\mathcal{DS}}_\pi(x)$, which is the fixed-point solution of

$$V(x) = T_\pi^{DS}[V](x), \quad \forall x \in \mathcal{X}'.$$

By definition, the following properties always hold:

$$\overline{\mathcal{DS}}_\pi(x) \ge d_0, \quad \overline{\mathcal{DS}}_\pi(x) \ge \mathcal{DS}_\pi(x) \ge \mathcal{D}_\pi(x), \text{ for any } x \in \mathcal{X}'.$$

To understand how this surrogate value function guarantees feasibility in problem $\mathcal{OPT}$, consider the optimal value function

$$\overline{\mathcal{DS}}(x) = \min_{\pi \in \Delta(x)} \overline{\mathcal{DS}}_\pi(x), \quad \forall x \in \mathcal{X}',$$

which is also the unique solution w.r.t. the fixed-point equation $\min_{\pi \in \Delta(x)} T_\pi^{DS}[V](x) = V(x)$. Now suppose at state $x_0$, the following condition holds: $\overline{\mathcal{DS}}(x_0) \le d_0$. Then there exists a policy $\pi$ that is is safe w.r.t. problem $\mathcal{OPT}$, i.e., $\mathcal{D}_\pi(x_0) \le \overline{\mathcal{DS}}_\pi(x_0) = d_0$.

Motivated by the above observation, we first check if the following condition holds:

$$\overline{\mathcal{DS}}(x_0) \le d_0.$$

If that is the case, define the set of feasible policies that is induced by the super-martingale $\overline{\mathcal{DS}}$ as

$$\mathcal{F}^{\text{DS}}(x) := \left\{ \pi \in \Delta : T_\pi^{DS}[\overline{\mathcal{DS}}](x) = \overline{\mathcal{DS}}(x) \right\},$$

and solve the following problem, whose solution (if exists) is guaranteed to be safe.

> **Problem $\mathcal{OPT}^{\text{DS}}$:** Assume $\overline{\mathcal{DS}}(x_0) \le d_0$. Then given an initial state $x_0 \in \mathcal{X}'$, and a threshold $d_0 \in \mathbb{R}_{\ge 0}$, solve $\min_{\pi \in \bigcup_{x \in \mathcal{X}'} \mathcal{F}^{\text{DS}}(x)} \mathbb{E}\left[ \sum_{t=0}^{\text{T}^*-1} c(x_t, a_t) \mid x_0, \pi \right]$.

Similar to the step-wise approach, clearly the $\overline{\mathcal{DS}}$-induced Bellman operator

$$T_{\mathcal{F}^{\text{DS}}}[V](x) = \min_{\pi \in \mathcal{F}^{\text{DS}}(d_0, x)} T_{\pi, c}[V](x)$$

is a contraction operator. This implies that there exists a unique solution $V_{\mathcal{F}^{\text{DS}}}^*$ to the fixed-point equation $T_{\mathcal{F}^{\text{DS}}}[V](x) = V(x)$, for $x \in \mathcal{X}'$, such that $V_{\mathcal{F}^{\text{DS}}}^*(x_0)$ is a solution to problem $\mathcal{OPT}^{\text{DS}}$.

# C Proofs of the Technical Results in Section 3

## C.1 Proof of Lemma 1

In the following part of the analysis we will use shorthand notation $P^*$ to denote the transition probability for $x \in \mathcal{X}'$ induced by the optimal policy, and $P_B$ to denote the transition probability for $x \in \mathcal{X}'$ induced by the baseline policy. These matrices are sub-stochastic because we exclude the terms in the recurrent states. This means that both spectral radii $\rho(P^*)$ and $\rho(P_B)$ are less than 1, and thus both $(I - P^*)$ and $(I - P_B)$ are invertible. By the Newmann series expansion, one can also show that

$$(I - P^*)^{-1} = \left\{ \sum_{t=0}^{\mathrm{T}^*-1} \mathbb{P}(x_t = x'|x_0 = x, \pi^*) \right\}_{x,x' \in \mathcal{X}'},$$

and

$$(I - P_B)^{-1} = \left\{ \sum_{t=0}^{\mathrm{T}^*-1} \mathbb{P}(x_t = x'|x_0 = x, \pi_B) \right\}_{x,x' \in \mathcal{X}'}.$$

We also define $\Delta(a|x) = \pi_B(a|x) - \pi^*(a|x)$ for any $x \in \mathcal{X}'$ and $a \in \mathcal{A}$, and $P_\Delta = \{\sum_{a \in \mathcal{A}} P(x'|x,a)\Delta(a|x)\}_{x,x' \in \mathcal{X}'}$. Therefore, one can easily see that

$$(I - P^*)(I - P_B + P_\Delta)^{-1} = I_{|\mathcal{X}'| \times |\mathcal{X}'|}.$$

Therefore, by the Woodbury Sherman Morrison identity, we have that

$$(I - P^*)^{-1} = (I - P_B)^{-1}(I_{|\mathcal{X}'| \times |\mathcal{X}'|} + P_\Delta(I - P^*)^{-1}). \tag{11}$$

By multiplying the constraint cost function vector $d(x)$ on both sides of the above equality, This further implies that for each $x \in \mathcal{X}'$, one has

$$\mathcal{D}_{\pi^*}(x) = \mathbb{E}\left[ \sum_{t=0}^{\mathrm{T}^*-1} d(x_t) + \epsilon(x_t) \mid \pi_B, x \right] = L_\epsilon(x), \tag{12}$$

such that

$$T_{\pi^*}[L_\epsilon](x) = L_\epsilon(x), \ \forall x \in \mathcal{X}'.$$

Here, the auxiliary constraint cost is given by

$$\epsilon(x) = \sum_{a \in \mathcal{A}} \Delta(a|x) \sum_{x' \in \mathcal{X}'} P(x'|x,a)\mathcal{D}_{\pi^*}(x'). \tag{13}$$

By construction, equation (11) immediately implies that $L_\epsilon$ is a fixed point solution of $T_{\pi^*}[V](x) = V(x)$ for $x \in \mathcal{X}'$. Furthermore, equation (13) further implies that the upper bound of the constraint cost $\epsilon$ is given by:

$$-2\overline{\mathrm{T}} D_{\max} D_{TV}(\pi^*||\pi_B)(x) \le \epsilon(x) \le 2\overline{\mathrm{T}} D_{\max} D_{TV}(\pi^*||\pi_B)(x), \ x \in \mathcal{X}',$$

where $\overline{\mathrm{T}}$ is the uniform upper-bound of the MDP stopping time.

Since $\pi^*$ is also a feasible policy of problem $\mathcal{OPT}$, this further implies that $L_\epsilon(x_0) \le d_0$.

## C.2 Proof of Theorem 1

First, under Assumption 1 the following inequality holds:

$$D_{TV}(\pi^*||\pi_B)(x) \le \frac{d_0 - \mathcal{D}^{\pi_B}(x_0)}{2\overline{\mathrm{T}}^2 D_{\max}}, \ \forall x \in \mathcal{X}'.$$

Recall that $\epsilon^*(x) = 2\overline{\mathrm{T}} D_{\max} D_{TV}(\pi^*||\pi_B)(x)$. The above expression implies that

$$\mathbb{E}\left[ \sum_{t=0}^{\mathrm{T}^*-1} \epsilon^*(x_t)|\pi_B, x_0 \right] \le 2\overline{\mathrm{T}}^2 D_{\max} \max_{x \in \mathcal{X}'} D_{TV}(\pi^*||\pi_B)(x) \le d_0 - \mathcal{D}^{\pi_B}(x_0),$$

which further implies

$$L_{\epsilon^*}(x_0) = \mathbb{E}\left[\sum_{t=0}^{\mathrm{T}^*-1} \epsilon^*(x_t)|\pi_B, x_0\right] + \mathcal{D}^{\pi_B}(x_0) \le d_0,$$

i.e., the second property in (2) holds.

Second, recall the following equality from (12):

$$(I - P^*)(I - P_B)^{-1}(\{d(x)\}_{x \in \mathcal{X}'} + \{\epsilon(x)\}_{x \in \mathcal{X}'}) = \{d(x)\}_{x \in \mathcal{X}'}$$

with the definition of the auxiliary constraint cost $\epsilon$ given by (13). We want to show that the first condition in (2) holds. By adding the term $(I - P^*)(I - P_B)^{-1}\epsilon^*$ to both sides of the above equality, it implies that:

$$(I - P^*)(I - P_B)^{-1}(D + \epsilon^*) = D + (I - P^*)(I - P_B)^{-1}(\epsilon^* - \epsilon),$$

where $\epsilon^*(x) - \epsilon(x) \ge 0$, for $x \in \mathcal{X}'$. Therefore, the proof is completed if we can show that for any $x \in \mathcal{X}'$:

$$\{(I - P^*)(I - P_B)^{-1}(\epsilon^* - \epsilon)\}(x) \ge 0. \tag{14}$$

Now consider the following inequalities derived from Assumption 1:

$$\max_x D_{TV}(\pi^*||\pi_B)(x) \le \frac{1}{2\overline{\mathrm{T}}}\frac{\overline{\mathrm{T}}D_{\max} - \overline{\mathcal{D}}}{\overline{\mathrm{T}}D_{\max} + \overline{\mathcal{D}}} \le \frac{1}{2\overline{\mathrm{T}}}\frac{\overline{\mathrm{T}}D_{\max} - \overline{\mathcal{D}}^*}{\overline{\mathrm{T}}D_{\max} + \overline{\mathcal{D}}^*},$$

the last inequality is due to the fact that $\overline{\mathcal{D}} \ge \overline{\mathcal{D}}^*$, where $\overline{\mathcal{D}}^* = \max_x \mathcal{D}_{\pi^*}(x)$ is the constraint upper-bound w.r.t. optimal policy $\pi^*$. Multiplying the ratio $D_{TV}(\pi^*||\pi_B)(x)/\max_x D_{TV}(\pi^*||\pi_B)(x) \ge 0$ on both sides of the above inequality, for each $x \in \mathcal{X}'$ one obtains the following inequality:

$$
\begin{aligned}
D_{TV}(\pi^*||\pi_B)(x) &\le \frac{(\overline{\mathrm{T}}D_{\max} - \overline{\mathcal{D}}^*)D_{TV}(\pi^*||\pi_B)(x)}{2\overline{\mathrm{T}}(\overline{\mathrm{T}}D_{\max} + \overline{\mathcal{D}}^*)\max_x D_{TV}(\pi^*||\pi_B)(x)} \\
&\le \frac{\epsilon^*(x) - \epsilon(x)}{2\overline{\mathrm{T}}\max_x\{\epsilon^*(x) - \epsilon(x)\}},
\end{aligned}
$$

the last inequality holds due to the fact that for any $x \in \mathcal{X}'$,

$$2(\overline{\mathrm{T}}D_{\max} - \overline{\mathcal{D}}^*)D_{TV}(\pi^*||\pi_B)(x) \le \epsilon^*(x) - \epsilon(x) \le 2(\overline{\mathrm{T}}D_{\max} + \overline{\mathcal{D}}^*)D_{TV}(\pi^*||\pi_B)(x).$$

Multiplying $2\overline{\mathrm{T}}\max_x\{\epsilon^*(x) - \epsilon(x)\}$ on both sides, it further implies that for each $x \in \mathcal{X}'$, one has the following inequality:

$$(\epsilon^*(x) - \epsilon(x)) - 2\overline{\mathrm{T}}\max_{x'}\{\epsilon^*(x') - \epsilon(x')\}D_{TV}(\pi^*||\pi_B)(x) \ge 0.$$

Now recall that $P^* = P_B - P_\Delta$, where $\Delta$ is equal to the matrix that characterizes the difference between the baseline and the optimal policy for each state in $\mathcal{X}'$ and action in $\mathcal{A}$, i.e., $\Delta(a|x) = \pi_B(a|x) - \pi^*(a|x), \forall x \in \mathcal{X}', \forall a \in \mathcal{A}$ and $P_\Delta = \{\sum_{a \in \mathcal{A}} P(x'|x, a)\Delta(a|x)\}_{x, x' \in \mathcal{X}'}$, the above condition guarantees that

$$\{(I - P_B + P_\Delta)(I - P_B)^{-1}(\epsilon^* - \epsilon)\}(x) \ge 0, \ \forall x \in \mathcal{X}'.$$

This finally comes to the conclusion that under Condition 1, the inequality in (14) holds, which further implies that

$$(I - P^*)(I - P_B)^{-1}(d(x) + \epsilon^*(x)) \ge d(x), \ \forall x \in \mathcal{X},$$

i.e., the first property in (2) holds with $L_{\epsilon^*}(x) = \mathbb{E}\left[\sum_{t=0}^{\mathrm{T}^*-1} d(x) + \epsilon^*(x)|\pi_B, x\right]$.

By combining the above results, one shows that $L_{\epsilon^*}$ is a Lyapunov function that satisfies the properties in (2) and (3), which concludes the proof.

## C.3 Properties of Safe Bellman Operator

**Proposition 3.** *The safe Bellman operator has the following properties.*

- ***Contraction***: *There exists a vector with positive components, i.e., $\rho : \mathcal{X} \to \mathbb{R}_{\geq 0}$, and a discounting factor $0 < \gamma < 1$ such that*

$$\|T[V] - T[W]\|_\rho \leq \gamma \|V - W\|_\rho,$$

*where the weighted norm is defined as $\|V\|_\rho = \max_{x \in \mathcal{X}} \frac{V(x)}{\rho(x)}$.*

- ***Monotonicity***: *For any value functions $V, W : \mathcal{X} \to \mathbb{R}$ such that $V(x) \leq W(x)$, one has the following inequality: $T[V](x) \leq T[W](x)$, for any state $x \in \mathcal{X}$.*

*Proof.* First, we show the monotonicity property. For the case of $x \in \mathcal{X} \setminus \mathcal{X}'$, the property trivially holds. For the case of $x \in \mathcal{X}'$, given value functions $W, V : \mathcal{X}' \to \mathbb{R}$ such that $V(x) \leq W(x)$ for any $x \in \mathcal{X}'$, by the definition of Bellman operator $T$, one can show that for any $x \in \mathcal{X}'$ and any $a \in \mathcal{A}$,

$$c(x, a) + \sum_{x' \in \mathcal{X}'} P(x'|x, a)V(x') \leq c(x, a) + \sum_{x' \in \mathcal{X}'} P(x'|x, a)W(x').$$

Therefore, by multiplying $\pi(a|x)$ on both sides, summing the above expression over $a \in \mathcal{A}$, and taking the minimum of $\pi$ over the feasible set $\mathcal{F}_{L_{\epsilon^*}}(x)$, one can show that $T[V](x) \leq T[W](x)$ for any $x \in \mathcal{X}'$.

Second we show that the contraction property holds. For the case of $x \in \mathcal{X} \setminus \mathcal{X}'$, the property trivially holds. For the case of $x \in \mathcal{X}'$, following the construction in Proposition 3.3.1 of [7], consider a stochastic shortest path problem where the transition probabilities and the constraint cost function are the same as the one in problem $\mathcal{OPT}$, but the cost are all equal to $-1$. Then, there exists a fixed point value function $\hat{V}$, such that

$$\hat{V}(x) = -1 + \min_{\pi \in \mathcal{F}_{L_{\epsilon^*}}(x)} \sum_a \pi(a|x) \sum_{x' \in \mathcal{X}'} P(x'|x, a)\hat{V}(x'), \ \forall x \in \mathcal{X}',$$

such that the following inequality holds for given feasible Markovian policy $\pi'$:

$$\hat{V}(x) \leq -1 + \sum_a \pi'(a|x) \sum_{x' \in \mathcal{X}'} P(x'|x, a)\hat{V}(x'), \ \forall x \in \mathcal{X}'.$$

Notice that $\hat{V}(x) \leq -1$ for all $x \in \mathcal{X}'$. By defining $\rho(x) = -\hat{V}(x)$, and by constructing $\gamma = \max_{x \in \mathcal{X}'}(\rho(x) - 1)/\rho(x)$, one immediately has $0 < \gamma < 1$, and

$$\sum_a \pi'(a|x) \sum_{x' \in \mathcal{X}'} P(x'|x, a)\rho(x') \leq \rho(x) - 1 \leq \gamma\rho(x), \ \forall x \in \mathcal{X}'.$$

Then by using Proposition 1.5.2 of [7], one can show that $T$ is a contraction operator. $\square$

## C.4 Proof of Theorem 2

Let $V_{\mathcal{OPT}}(x_0)$ be the optimal value function of problem $\mathcal{OPT}$, and let $V^*$ be a fixed point solution: $V(x) = T[V](x)$, for any $x \in \mathcal{X}$. For the case when $x_0 \in \mathcal{X} \setminus \mathcal{X}'$, the following result trivially holds: $V_{\mathcal{OPT}}(x_0) = T[V_{\mathcal{OPT}}](x_0) = V^*(x_0) = 0$. Below, we show the equality for the case of $x_0 \in \mathcal{X}'$.

First, we want to show that $V_{\mathcal{OPT}}(x_0) \leq V^*(x_0)$. Consider the greedy policy $\bar{\pi}^*$ constructed from the fixed point equation. Immediately, one has that $\bar{\pi}^*(\cdot|x) \in \mathcal{F}_{L_{\epsilon^*}}(x)$. This implies

$$T_{\bar{\pi}^*, d}[L_{\epsilon^*}](x) \leq L_{\epsilon^*}(x), \ \forall x \in \mathcal{X}'. \tag{15}$$

Thus by recursively applying $T_{\bar{\pi}^*, d}$ on both sides of the above inequality, the contraction property of Bellman operator $T_{\bar{\pi}^*, d}$ implies that one has the following expression:

$$\lim_{n \to \infty} T_{\bar{\pi}^*, d}^n[L_{\epsilon^*}](x_0) = \mathbb{E}\left[\sum_{t=0}^{\mathrm{T}^*-1} d(x_t) + L_{\epsilon^*}(x_{\mathrm{T}^*}) \mid x_0, \bar{\pi}^*\right] \leq L_{\epsilon^*}(x_0) \leq d_0.$$

Since the state enters the terminal set at time $t = \mathrm{T}^*$, we have that $L_{\epsilon^*}(x_{\mathrm{T}^*}) = 0$ almost surely. Then the above inequality implies $\mathbb{E}\left[\sum_{t=0}^{\mathrm{T}^*-1} d(x_t) \mid x_0, \overline{\pi}^*\right] \leq d_0$, which further shows that $\overline{\pi}^*$ is a feasible policy to problem $\mathcal{OPT}$. On the other hand, recall that $V^*(x)$ is a fixed point solution to $V(x) = T[V](x)$, for any $x \in \mathcal{X}'$. Then for any bounded initial value function $V_0$, the contraction property of Bellman operator $T_{\overline{\pi}^*,c}$ implies that

$$V^*(x) = \lim_{n\to\infty} T_{\overline{\pi}^*,c}^n[V_0](x) = \lim_{n\to\infty} \mathbb{E}\left[\sum_{t=0}^{n-1} c(x_t, a_t) + V_0(x_n) \mid x_0 = x, \overline{\pi}^*\right],$$

for which the transient assumption of stopping MDPs further implies that

$$V^*(x) = \mathbb{E}\left[\sum_{t=0}^{\mathrm{T}^*-1} c(x_t, a_t) \mid x_0 = x, \overline{\pi}^*\right].$$

Since $\overline{\pi}^*$ is a feasible solution to problem $\mathcal{OPT}$. This further implies that $V_{\mathcal{OPT}}(x_0) \leq V^*(x_0)$.

Second, we want to show that $V_{\mathcal{OPT}}(x_0) \geq V^*(x_0)$. Consider the optimal policy $\pi^*$ of problem $\mathcal{OPT}$ that is used to construct Lyapunov function $L_{\epsilon^*}$. Since the Lyapunov function satisfies the following Bellman inequality:

$$T_{\pi^*,d}[L_{\epsilon^*}](x) \leq L_{\epsilon^*}(x), \ \forall x \in \mathcal{X}',$$

it implies that the optimal policy $\pi^*$ is a feasible solution to the optimization problem in Bellman operator $T[V^*](x)$. Note that $V^*$ is a fixed point solution to equation: $V^*(x) = T[V^*](x)$, for any $x \in \mathcal{X}'$. Immediately the above result yields the following inequality:

$$V^*(x) = T_{\overline{\pi}^*,c}[V^*](x) \leq T_{\pi^*,c}[V^*](x), \ \forall x \in \mathcal{X}',$$

the first equality holds because $\overline{\pi}^*(\cdot|x)$ is the minimizer of the optimization problem in $T[V^*](x)$, $x \in \mathcal{X}'$. By recursively applying Bellman operator $T_{\pi^*,c}$ to this inequality, one has the following result:

$$V^*(x) \leq \lim_{n\to\infty} T_{\pi^*,c}^n[V^*](x) = \mathcal{C}_{\pi^*}(x) = V_{\mathcal{OPT}}(x), \ \forall x \in \mathcal{X}'.$$

One thus concludes that $V_{\mathcal{OPT}}(x_0) \geq V^*(x_0)$.

Combining the above analysis, we prove the claim of $V_{\mathcal{OPT}}(x_0) = V^*(x_0)$, and the greedy policy of the fixed-point equation, i.e., $\overline{\pi}^*$, is an optimal policy to problem $\mathcal{OPT}$.

### C.5 Proof of Proposition 1

For the derivations of consistent feasibility and policy improvement, without loss of generality we only consider the case of $k = 0$.

To show the property of consistent feasibility, consider an arbitrary feasible policy $\pi_0$ of problem $\mathcal{OPT}$. By definition, one has $\mathcal{D}_{\pi_0}(x_0) \leq d_0$, and the value function $\mathcal{D}_{\pi_0}$ has the following property:

$$\sum_a \pi_0(a|x)\left[d(x) + \sum_{x'} P(x'|x, a)\mathcal{D}_{\pi_0}(x')\right] = \mathcal{D}_{\pi_0}(x), \ \forall x \in \mathcal{X}'.$$

Immediately, since $\mathcal{D}_{\pi_0}$ satisfies the constraint in (5), one can treat it as a Lyapunov function, this shows that the set of Lyapunov functions $\mathcal{L}_{\pi_0}(x_0, d_0)$ is non-empty. Therefore, there exists a bounded Lyapunov function $\{L_{\epsilon_0}(x)\}_{x\in\mathcal{X}}$ as the solution of the optimization problem in Step 0. Now consider the policy optimization problem in Step 1. Based on the construction of $\{L_{\epsilon_0}(x)\}_{x\in\mathcal{X}}$, the current policy $\pi_0$ is a feasible solution to this problem, therefore the feasibility set is non-empty. Furthermore, by recursively applying the inequality constraint on the updated policy $\pi_1$ for $\mathrm{T}^* - 1$ times, one has the following inequality:

$$\mathbb{E}\left[\sum_{t=0}^{\mathrm{T}^*-1} d(x_t) + L_{\epsilon_0}(x_{\mathrm{T}^*})|x_0, \pi_1\right] \leq L_{\epsilon_0}(x_0) \leq d_0.$$

This shows that $\pi_1$ is a feasible policy to problem $\mathcal{OPT}$.

To show the property of policy improvement, consider the policy optimization in Step 1. Notice that the current policy $\pi_0$ is a feasible solution of this problem (with Lyapunov function $L_0$), and the updated policy $\pi_1$ is a minimizer of this problem. Then, one immediately has the following chain of inequalities:

$$T_{\pi_1,c}[V_0](x) = \sum_{a \in \mathcal{A}} \pi_1(a|x) \left[ c(x,a) + \sum_{x' \in \mathcal{X}'} P(x'|x,a) V_0(x') \right]$$

$$\leq \sum_{a \in \mathcal{A}} \pi_0(a|x) \left[ c(x,a) + \sum_{x' \in \mathcal{X}'} P(x'|x,a) V_0(x') \right] = V_0(x), \ \forall x \in \mathcal{X}',$$

where the last equality is due to the fact that $V_0(x) = \mathcal{C}_{\pi_0}(x)$, for any $x \in \mathcal{X}$. By the contraction property of Bellman operator $T_{\pi_1}$, the above condition further implies

$$\mathcal{C}_{\pi_1}(x) = \lim_{n \to \infty} T^n_{\pi_1,c}[V_0](x) \leq V_0(x) = \mathcal{C}_{\pi_0}(x), \ \forall x \in \mathcal{X}',$$

which proves the claim about policy improvement.

To show the property of asymptotic convergence, notice that the value function sequence $\{\mathcal{C}_{\pi_k}(\cdot)\}_{k \geq 0}$ is uniformly monotonic, and each element is uniformly lower bounded by the unique solution of fixed point equation: $V(x) = \min_{a \in \mathcal{A}} c(x,a) + \sum_{x' \in \mathcal{X}'} P(x'|x,a) V(x')$, $\forall x \in \mathcal{X}'$. Therefore, this sequence of value function converges (point-wise) as soon as in the limit the policy improvement stops. Whenever this happens, i.e., there exists $K \geq 0$ such that $\max_{x \in \mathcal{X}'} |\mathcal{C}_{\pi_{K+1}}(x) - \mathcal{C}_{\pi_K}(x)| \leq \epsilon$ for any $\epsilon > 0$, then this value function is the fixed point of $\min_{\pi \in \mathcal{F}_{L_K}(x)} T_{\pi,c}[V](x) = V(x)$, $\forall x \in \mathcal{X}'$, whose solution policy is unique (due to the strict convexity of the objective function in the policy optimization problem after adding a convex regularizer). Furthermore, due to the strict concavity of the objective function in problem in (5) (after adding a concave regularizer), the solution pair of this problem is unique, which means the update of $\{(L_{\epsilon_k}, \epsilon_k)\}$ stops at step $K$. Together, this also means that policy update $\{\pi_k\}$ converges.

### C.6 Analysis on Performance Improvement in Safe Policy Iteration

Similar to the analysis in [2], the following lemma provides a bound in policy improvement.

**Lemma 2.** *For any policies $\pi'$ and $\pi$, define the following error function:*

$$\Lambda(\pi, \pi') = \mathbb{E}\left[ \sum_{t=0}^{T^*-1} \left( \frac{\pi'(a_t|x_t)}{\pi(a_t|x_t)} - 1 \right) \underbrace{(C(x_t, a_t) + V^\pi(x_{t+1}) - V^\pi(x_t))}_{\delta^\pi(x_t, a_t, x_{t+1})} \mid x_0, \pi \right]$$

$$= \mathbb{E}\left[ \sum_{t=0}^{T^*-1} \mathbb{E}_{a \sim \pi'(\cdot|x)}[Q^\pi(x_t, a)] - V^\pi(x_t) \mid x_0, \pi \right]$$

*and $\Delta^{\pi'} = \max_{x \in \mathcal{X}'} |\mathbb{E}_{a \sim \pi'(\cdot|x)}[Q^\pi(x, a)] - V^\pi(x)|$. Then, the following error bound on the performance difference between $\pi$ and $\pi'$ holds:*

$$\Lambda(\pi, \pi') - \mathcal{E}^{\pi,\pi'}_{TV} \leq \mathcal{C}_{\pi'}(x_0) - \mathcal{C}_\pi(x_0) \leq \Lambda(\pi, \pi') + \mathcal{E}^{\pi,\pi'}_{TV}.$$

*where $\mathcal{E}^{\pi,\pi'}_{TV} = 2\Delta^{\pi'} \cdot \left( \max_{x_0 \in \mathcal{X}'} \mathbb{E}[T^*|x_0, \pi'] \right) \cdot \mathbb{E}\left[ \sum_{t=0}^{T^*-1} D_{TV}(\pi'||\pi)(x_t) \mid x_0, \pi \right]$.*

*Proof.* First, it is clear from the property of telescopic sum that

$$\mathcal{C}_{\pi'}(x_0) - \mathcal{C}_\pi(x_0) = \mathbb{E}\left[ \sum_{t=0}^{T^*-1} \delta^{\pi'}(x_t, a_t, x_{t+1}) \mid x_0, \pi' \right] - \mathbb{E}\left[ \sum_{t=0}^{T^*-1} \delta^\pi(x_t, a_t, x_{t+1}) \mid x_0, \pi \right]$$

$$\leq \mathbb{E}\left[ \sum_{t=0}^{T^*-1} \delta^{\pi'}(x_t, a_t, x_{t+1}) - \delta^\pi(x_t, a_t, x_{t+1}) \mid x_0, \pi \right] + \Delta^{\pi'} \cdot \sum_{y \in \mathcal{X}'} \left| \sum_{t=0}^{T^*-1} \mathbb{P}(x_t = y|x_0, \pi') - \sum_{t=0}^{T^*-1} \mathbb{P}(x_t = y|x_0, \pi) \right|,$$

where the inequality is based on the Holder inequality $\mathbb{E}[|xy|] \leq \left(\mathbb{E}[|x|^p]\right)^{1/p} \left(\mathbb{E}[|y|^q]\right)^{1/q}$ with $p = 1$ and $q = \infty$.

Immediately, the first part of the above expression is re-written as: $\mathbb{E}\left[\sum_{t=0}^{\mathrm{T}^*-1} \delta^{\pi'}(x_t, a_t, x_{t+1}) - \delta^\pi(x_t, a_t, x_{t+1}) \mid x_0, \pi\right] = \Lambda(\pi, \pi')$. Recall that shorthand notation $P_\pi$ to denote the transition probability for $x \in \mathcal{X}'$ induced by the policy $\pi$. For the second part of the above expression, notice that the following chain of inequalities holds:

$$\sum_{y \in \mathcal{X}'} \left| \sum_{t=0}^{\mathrm{T}^*-1} \mathbb{P}(x_t = y | x_0, \pi) - \sum_{t=0}^{\mathrm{T}^*-1} \mathbb{P}(x_t = y | x_0, \pi') \right|$$

$$= \sum_{y \in \mathcal{X}'} \left| \mathbf{1}(x_0)^\top \left( (I - P_\pi)^{-1} - (I - P_{\pi'})^{-1} \right) \mathbf{1}(y) \right|$$

$$= \sum_{y \in \mathcal{X}'} \left| \mathbf{1}(x_0)^\top (I - P_\pi)^{-1}(P_\pi - P_{\pi'})(I - P_{\pi'})^{-1}\mathbf{1}(y) \right|$$

$$\leq \|(\mathbf{1}(x_0)^\top (I - P_\pi)^{-1}(P_\pi - P_{\pi'})\|_1 \cdot \sum_{y \in \mathcal{X}'} \max_{x_0 \in \mathcal{X}'} \mathbf{1}(x_0)(I - P_{\pi'})^{-1}\mathbf{1}(y)$$

$$\leq \sum_{y \in \mathcal{X}'} \overline{\mathrm{T}} \cdot \mathbf{1}(y) \cdot \mathbb{E}\left[ \sum_{t=0}^{\mathrm{T}^*-1} \sum_{x' \in \mathcal{X}'} P(x'|x_t, a_t) \sum_{a \in \mathcal{A}} |\pi(a_t|x_t) - \pi'(a_t|x_t)| \mid x_0, \pi \right]$$

$$= 2\overline{\mathrm{T}} \cdot \mathbb{E}\left[ \sum_{t=0}^{\mathrm{T}^*-1} D_{TV}(\pi'\|\pi)(x_t) \mid x_0, \pi \right],$$

the first, is based on the Holder inequality with $p = 1$ and $q = \infty$ and on the fact that all entries in $(I - P_{\pi'})^{-1}$ is non-negative, the second inequality is due to the fact that starting at any initial state $x_0$, it almost takes $\overline{\mathrm{T}}$ steps to the set of recurrent states $\mathcal{X} \setminus \mathcal{X}'$. In other words, one has the following inequality:

$$\mathbf{1}(x_0)(I - P_{\pi'})^{-1}\mathbf{1}(y) = \mathbb{E}\left[ \sum_{t=0}^{\mathrm{T}^*-1} \mathbf{1}\{x_t = y\} \mid x_0, \pi' \right] \leq \overline{\mathrm{T}}, \ \forall x_0, y \in \mathcal{X}'.$$

Therefore, combining with these properties the proof of the above error bound is completed. $\square$

Using this result, the sub-optimality performance bound of policy $\pi_{k^*}$ from SPI is $\overline{\mathrm{T}}C_{\max} - \left(\sum_{k=0}^{k^*-1} \max\{0, \Lambda^{\pi_k, \pi_{k+1}} - \mathcal{E}_{TV}^{\pi_k, \pi_{k+1}}\} + \mathcal{C}_{\pi_0}(x_0)\right)$.

### C.7 Proof of Proposition 2

For the derivations of consistent feasibility and monotonic improvement on value estimation, without loss of generality we only consider the case of $t = 0$.

To show the property of consistent feasibility, notice that with the definitions of the initial $Q$-function $Q_0$, the initial Lyapunov function $L_{\epsilon_0}$ w.r.t. the initial auxiliary cost $\epsilon_0$, the corresponding induced policy $\pi_0$ is feasible to problem $\mathcal{OPT}$, i.e., $\mathcal{D}_{\pi_0}(x_0) \leq d_0$. Consider the optimization problem in Step 1. Immediately, since $\mathcal{D}_{\pi_0}$ satisfies the constraint in (5), one can treat it as a Lyapunov function, and the set of Lyapunov functions $\mathcal{L}_{\pi_0}(x_0, d_0)$ is non-empty. Therefore, there exists a bounded Lyapunov function $\{L_{\epsilon_1}(x)\}_{x \in \mathcal{X}}$ and auxiliary cost $\{\epsilon_1(x)\}_{x \in \mathcal{X}}$ as the solution of this optimization problem. Now consider the policy update in Step 0. Since $\pi_1(\cdot|\cdot)$ belongs to the set of feasible policies $\mathcal{F}_{L_{\epsilon_1}}(\cdot)$, by recursively applying the inequality constraint on the updated policy $\pi_1$ for $\mathrm{T}^* - 1$ times, one has the following inequality:

$$\mathbb{E}\left[ \sum_{t=0}^{\mathrm{T}^*-1} d(x_t) + L_{\epsilon_1}(x_{\mathrm{T}^*})|x_0, \pi_1 \right] \leq L_{\epsilon_1}(x_0) \leq d_0.$$

This shows that $\pi_1$ is a feasible policy to problem $\mathcal{OPT}$.

To show the asymptotic convergence property, for every initial state $x_0$ and any time step $K$, with the policy $\pi = \{\pi_0, \ldots, \pi_{K-1}\}$ generated by the value iteration procedure, the cumulative cost can be broken down into the following two portions, which consists of the cost over the first $K$ stages and the remaining cost. Specifically,

$$\overline{V}(x_0) = \mathbb{E}\left[\sum_{t=0}^{\mathrm{T}^*-1} c(x_t, a_t) \mid x_0, \pi\right] = \mathbb{E}\left[\sum_{t=0}^{K-1} c(x_t, a_t) \mid x_0, \pi\right] + \mathbb{E}\left[\sum_{t=K}^{\mathrm{T}^*-1} c(x_t, a_t) \mid x_0, \pi\right],$$

where the second term is bounded $\mathbb{E}[\mathrm{T}^* - K \mid \pi, x_0] C_{\max}$, which is bounded by $\sum_{t=K}^{\infty} \mathbb{P}(x_t \in \mathcal{X}' \mid x_0, \pi) \cdot C_{\max} > 0$. Since the value function $V_0(x) = \min_{\pi \in \mathcal{F}_{L_{\epsilon_0}}(x)} \pi(\cdot|x)^\top Q_0(x, \cdot)$ is also bounded, one can further show the following inequality:

$$-\mathbb{P}(x_K \in \mathcal{X}' \mid x_0, \pi) \cdot \|V_0\|_\infty \leq T_{K-1}[\cdots [T_0[V_0]] \cdots](x_0) - \mathbb{E}\left[\sum_{t=0}^{K-1} c(x_t, a_t) \mid x_0, \pi\right]$$

$$\leq \mathbb{P}(x_K \in \mathcal{X}' \mid x_0, \pi) \cdot \|V_0\|_\infty.$$

Recall from our problem setting that all policies are proper (see Assumption 3.1.1 and Assumption 3.1.2 in [7]). Then by the property of a transient MDP (see Definition 7.1 in [3]), the sum of probabilities of the state trajectory after step $K$ that is in the transient set $\mathcal{X}'$, i.e., $\sum_{t=K}^{\infty} \mathbb{P}(x_t \in \mathcal{X}' \mid x_0, \pi)$, is bounded by $M_\pi \cdot \epsilon$. Therefore, as $K$ goes to $\infty$, $\epsilon$ approaches 0. Using the result that $\sum_{t=K}^{\infty} \mathbb{P}(x_t \in \mathcal{X}' \mid x_0, \pi)$ vanishes as $K$ goes to $\infty$, one concludes that

$$\lim_{K \to \infty} T_{K-1}[\cdots [T_0[V_0]] \cdots](x_0) = \overline{V}(x_0), \quad \forall x_0 \in \mathcal{X}',$$

which completes the proof of this property.

# D  Pseudo-code of Safe Reinforcement Learning Algorithms

---

**Algorithm 3** Policy Distillation

---

**Input:** Policy parameterization $\pi_\phi$ with parameter $\phi$; A batch of state trajectories $\{x_{0,m}, \ldots, x_{\overline{T}-1,m}\}_{m=1}^M$ generated by following the baseline policy $\pi_B$

Compute the action probabilities $\{\pi'(\cdot|x_{0,m}), \ldots, \pi'(\cdot|x_{\overline{T}-1,m})\}_{m=1}^M$ by solving problem in (16).

Compute the policy parameter by supervised learning:

$$\phi^* \in \arg\min_\phi \frac{1}{m} \sum_{m=1}^M \sum_{t=0}^{\overline{T}-1} D_{\text{JSD}}(\pi_\phi(\cdot|x_{t,m}) \parallel \pi'(\cdot|x_{t,m}))$$

where $D_{\text{JSD}}(P||Q) = \frac{1}{2}D(P \parallel \frac{1}{2}(P+Q)) + \frac{1}{2}D(Q \parallel \frac{1}{2}(P+Q))$ is the Jensen-Shannon divergence

**Return** Distilled policy $\pi_{\phi^*}$

---

**Algorithm 4** Safe DQN

---

**Input:** Initial prioritized replay buffer $M = \{\emptyset\}$; Initial importance weights $w_0 = 1$, $w_{D,0} = 1$ $w_{T,0} = 1$; Mini-batch size $|B|$; Network parameters $\theta^-$, $\theta_D^-$, $\theta_T^-$; Initial feasible policy $\pi_0$;

**for** $k \in \{0, 1, \ldots, \}$ **do**

    **for** $t = 0$ **to** $\overline{T} - 1$ **do**

        Sample action $a_t \sim \pi_k(\cdot|x_t)$, execute $a_t$ and observe next state $x_{t+1}$, and costs $(c_t, d_t)$

        Add experiences to replay buffer, i.e., $M \leftarrow (x_t, a_t, c_t, d_t, x_{t+1}, w_0, w_{D,0}, w_{T,0}) \cup M$

        From the buffer $M$, sample a mini-batch $B = \{(x_j, a_j, c_j, d_j, x'_j, w_j, w_{D,j}, w_{T,j})\}_{j=1}^{|B|}$ and set the targets $y_{D,j}$, $y_{T,j}$, and $y_j$

        Update the $\theta$ parameters as follows:

$$\theta_D \leftarrow \theta_D^- - \kappa_j \cdot w_{D,j} \cdot (y_{D,j} - \widehat{Q}_D(x_j, a_j; \theta_D^-)) \cdot \nabla_\theta \widehat{Q}_D(x_j, a_j; \theta)|_{\theta = \theta_D^-},$$

$$\theta_T \leftarrow \theta_T^- - \kappa_j \cdot w_{T,j} \cdot (y_{T,j} - \widehat{Q}_T(x_j, a_j; \theta_T^-)) \cdot \nabla_\theta \widehat{Q}_T(x_j, a_j; \theta)|_{\theta = \theta_T^-},$$

$$\theta \leftarrow \theta^- - \eta_j \cdot w_j \cdot (y_j - \widehat{Q}(x_j, a_j; \theta^-)) \cdot \nabla_\theta \widehat{Q}(x_j, a_j; \theta)|_{\theta = \theta^-}$$

        where the target values are respectively $y_{D,j} = d(x_j) + \pi_k(\cdot|x'_j)^\top \widehat{Q}_D(x'_j, \cdot; \theta_D)$, $y_{T,j} = \mathbf{1}\{x_j \in \mathcal{X}'\} + \pi_k(\cdot|x'_j)^\top \widehat{Q}_T(x'_j, \cdot; \theta_T)$, and $y_j = c(x_j, a_j) + \pi'(\cdot|x'_j)^\top \widehat{Q}(x'_j, \cdot; \theta)$, and $\pi'(\cdot|x'_j)$ is the greedy action probability w.r.t. $x'_j$, which is a solution to (6)

        Prioritized Sweep: Update importance weights $w_j$, $w_{D,j}$, and $w_{T,j}$ of the samples in mini-batch $B$, based on TD errors $|y_j - \widehat{Q}(x_j, a_j; \theta)|$, $|y_{D,j} - \widehat{Q}_D(x_j, a_j; \theta_D)|$ and $|y_{T,j} - \widehat{Q}_T(x_j, a_j; \theta_T)|$

        Distillation: Update the policy to $\pi_{k+1}$ using Algorithm 3 w.r.t. data $\{x'_{0,j}, \ldots, x'_{\overline{T}-1,j}\}_{j=1}^{|B|}$ and $\{\pi'(\cdot|x'_{0,j}), \ldots, \pi'(\cdot|x'_{\overline{T}-1,j})\}_{j=1}^{|B|}$

    **end for**

    Double $Q-$learning: Update $\theta^- = \theta$, $\theta_D^- = \theta_D$ and $\theta_T^- = \theta_T$ after $c$ iterations

**end for**

**Return**

---

---

**Algorithm 5** Safe DPI

---

**Input:** Initial prioritized replay buffer $M = \{\emptyset\}$; Initial importance weights $w_0 = 1$, $w_{D,0} = 1$, $w_{T,0} = 1$; Mini-batch size $|B|$; Initial feasible policy $\pi_0$;

**for** $k \in \{0, 1, \dots,\}$ **do**

    Sample action $a_t \sim \pi_k(\cdot|x_t)$, execute $a_t$ and observe next state $x_{t+1}$, and costs $(c_t, d_t)$

    Add experiences to replay buffer, i.e., $M \leftarrow (x_t, a_t, c_t, d_t, x_{t+1}, w_0, w_{D,0}, w_{T,0}) \cup M$

    From the buffer $M$, sample a mini-batch $B = \{(x_j, a_j, c_j, d_j, x'_j, w_j, w_{D,j}, w_{T,j})\}_{j=1}^{|B|}$ and set the targets $y_{D,j}$, $y_{T,j}$, and $y_j$

    Update the $\theta$ parameters as follows:

$$\theta^*_{\pi_k} \in \arg\min_\theta \sum_{t,j} \left(c_{t,j} - \widehat{Q}(x_{t,j}, a_{t,j}; \theta) + \pi_k(\cdot|x_{t+1,j})^\top \widehat{Q}(x_{t+1,j}, \cdot; \theta)\right)^2,$$

$$\theta^*_{D,\pi_k} \in \arg\min_{\theta_D} \sum_{t,j} \left(d_{t,j} - \widehat{Q}_D(x_{t,j}, a_{t,j}; \theta_D) + \pi_k(\cdot|x_{t+1,j})^\top \widehat{Q}_D(x_{t+1,j}, \cdot; \theta_D)\right)^2,$$

$$\theta^*_{T,\pi_k} \in \arg\min_{\theta_T} \sum_{t,j} \left(\mathbf{1}\{x_{t,j} \in \mathcal{X}'\} - Q_T(x_{t,j}, a_{t,j}; \theta_T) + \pi_k(\cdot|x_{t+1,j})^\top Q_T(x_{t+1,j}, \cdot; \theta_T)\right)^2$$

    and construct the $Q-$functions $\widehat{Q}(x, a; \theta^*_{\pi_k})$ and $\widehat{Q}_L(x, a; \theta^*_{D,\pi_k}, \theta^*_{T,\pi_k}) = \widehat{Q}_D(x, a; \theta_D) + \widetilde{\epsilon}' \cdot \widehat{Q}_T(x, a; \theta_T)$

    Calculate greedy action probabilities $\{\pi'(\cdot|x_{0,j}), \dots, \pi'(\cdot|x_{\overline{T}-1,j})\}_{j=1}^{|B|}$ by solving problem (6), with respect to batch of states drawn from the replay buffer $\{x_{0,j}, \dots, x_{\overline{T}-1,j}\}_{j=1}^{|B|}$

    Distillation: Update the policy to $\pi_{k+1}$ using Algorithm 3 w.r.t. data $\{x_{0,j}, \dots, x_{\overline{T}-1,j}\}_{j=1}^{|B|}$ and $\{\pi'(\cdot|x_{0,j}), \dots, \pi'(\cdot|x_{\overline{T}-1,j})\}_{j=1}^{|B|}$

**end for**

**Return** Final policy $\pi_{k^*}$

---

# E  Practical Implementations

There are several techniques that improve training and scalability of the safe RL algorithms. To improve stability in training $Q$ networks, one may apply double $Q-$learning [42] to separate the target values and the value function parameters and to slowly update the target $Q$ values at every predetermined iterations. On the other hand, to incentivize learning at state-action pairs that have high temporal difference (TD) errors, one can use a prioritized sweep in replay buffers [37] to add an importance weight to relevant experience. To extend the safe RL algorithms to handle continuous actions, one may adopt the normalized advantage functions (NAFs) [17] parameterization for $Q-$functions. Finally, instead of exactly solving the LP problem for policy optimization in (6), one may approximate this solution by solving its entropy regularized counterpart [29]. This approximation has an elegant closed-form solution that is parameterized by a Lagrange multiplier, which can be effectively computed by binary search methods (see Appendix E.1 and Appendix E.2 for details).

## E.1  Case 1: Discrete Action Space

In this case, problem (6) is cast as finite dimensional linear programming (LP). In order to effectively approximate the solution policy especially when the action space becomes large, instead of exactly solving this inner optimization problem, one considers its Shannon entropy-regularized variant:

$$\min_{\pi \in \Delta} \pi(\cdot|x)^\top \left(Q(x, \cdot) + \tau \log \pi(\cdot|x)\right) \tag{16}$$

$$\text{s.t. } (\pi(\cdot|x) - \pi_B(\cdot|x))^\top Q_L(x, \cdot) \leq \widetilde{\epsilon}'(x)$$

where $\tau > 0$ is the regularization constant. When $\tau \to 0$, then $\pi^*_\tau$ converges to the original solution policy $\pi^*$.

We will hereby illustrate how to effectively solve $\pi^*_\tau$ for any given $\tau > 0$ without explicitly solving the LP. Consider the Lagrangian problem for entropy-regularized optimization:

$$\min_{\pi \in \Delta} \max_{\lambda \geq 0} \Gamma_x(\pi, \lambda),$$

where $\Gamma_x(\pi, \lambda) = \pi(\cdot|x)^\top (Q(x, \cdot) + \lambda Q_L(x, \cdot) + \tau \log \pi(\cdot|x)) - \lambda (\pi_B(\cdot|x)^\top Q_L(x, \cdot) + \widetilde{\epsilon}'(x))$ is the Lagrangian function. Notice that the set of stationary Markovian policies $\Delta$ is a convex set, and the objective function is a convex function in $\pi$ and concave in $\lambda$. By strong duality, there exists a saddle-point to the Lagrangian problem where solution policy is equal to $\pi_\tau^*$, and it can be solved by the maximin problem:

$$\max_{\lambda \geq 0} \min_{\pi \in \Delta} \Gamma_x(\pi, \lambda).$$

For the inner minimization problem, it has been shown that the $\lambda-$solution policy has the following closed form:

$$\pi_{\tau, \lambda}^*(\cdot|x) \propto \exp\left(-\frac{Q(x, \cdot) + \lambda Q_L(x, \cdot)}{\tau}\right).$$

Equipped with this formulation, we now solve the problem for the optimal Lagrange multiplier $\lambda^*(x)$ at state $x \in \mathcal{X}'$:

$$\max_{\lambda \geq 0} -\tau \cdot \text{logsumexp}\left(-\frac{Q(x, \cdot) + \lambda Q_L(x, \cdot)}{\tau}\right) - \lambda (\pi_B(\cdot|x)^\top Q_L(x, \cdot) + \widetilde{\epsilon}'(x)),$$

where $\text{logsumexp}(y) = \log \sum_a \exp(y_a)$ is a strictly convex function in $y$, and the objective function is a concave function of $\lambda$. Notice that this problem has a unique optimal Lagrange multiplier that is the solution of the following KKT condition:

$$\pi_B(\cdot|x)^\top Q_L(x, \cdot) + \widetilde{\epsilon}'(x) + \frac{\sum_a Q_L(x, a) \exp\left(-\frac{Q(x, a) + \lambda Q_L(x, a)}{\tau}\right)}{\sum_a \exp\left(-\frac{Q(x, a) + \lambda Q_L(x, a)}{\tau}\right)} = 0.$$

Using the parameterization $z = \exp(-\lambda)$, this condition can be written as the following polynomial equation in $z$:

$$\sum_a \left(Q_L(x, a) + \pi_B(\cdot|x)^\top Q_L(x, \cdot) + \widetilde{\epsilon}'(x)\right) \cdot \exp\left(-\frac{Q(x, a)}{\tau}\right) \cdot z^{\frac{Q_L(x, a)}{\tau}} = 0. \qquad (17)$$

Therefore, the solution $0 \leq z^*(x) \leq 1$ can be solved by computing the root solution of the above polynomial and the optimal Lagrange multiplier is given by $\lambda^*(x) = -\log(z^*(x)) \geq 0$.

Combining the above results, the optimal policy of the entropy-regularized problem is therefore given by

$$\pi_\tau^*(\cdot|x) \propto \exp\left(-\frac{Q(x, \cdot) + \lambda^*(x) \cdot Q_L(x, \cdot)}{\tau}\right). \qquad (18)$$

### E.2  Case 2: Continuous Action Space

In order to effectively solve the inner optimization problem in (6) when the action space is continuous, on top of the using the entropy-regularized inner optimization problem in (16), we adopt the idea from the normalized advantage functions (NAF) approach for function approximation, where we express the $Q-$function and the state-action Lyapunov function with their second order Taylor-series expansions at an arbitrary action $\mu(x)$ as follows:

$$Q(x, a) \approx Q(x, \mu(x)) + \nabla_a Q(x, a)|_{a=\mu(x)} \cdot (a - \mu(x))$$
$$+ \frac{1}{2}(a - \mu(x))^\top \nabla_a^2 Q(x, a)|_{a=\mu(x)} (a - \mu(x)) + o(\|a - \mu(x)\|^3),$$
$$Q_L(x, a) \approx Q_L(x, \mu(x)) + \nabla_a Q_L(x, a)|_{a=\mu(x)} \cdot (a - \mu(x))$$
$$+ \frac{1}{2}(a - \mu(x))^\top \nabla_a^2 Q_L(x, a)|_{a=\mu(x)} (a - \mu(x)) + o(\|a - \mu(x)\|^3).$$

While these representations are more restrictive than the general function approximations, they provide a receipe to determine the policy, which is a minimizer of problem (6) analytically for the updates in the safe AVI and safe DQN algorithms. In particular, using the above parameterizations, notice that

$$\text{logsumexp}\left(-\frac{Q(x, \cdot) + \lambda^*(x) \cdot Q_L(x, \cdot)}{\tau}\right)$$
$$= -\frac{1}{2} \log |A_{\lambda^*}(x)| - \frac{1}{2}\psi_{\lambda^*}(x)^\top A_{\lambda^*}^{-1}(x) \psi_{\lambda^*}(x) + K_{\lambda^*}(x) + \frac{n}{2}\log(2\pi),$$

where $n$ is the dimension of actions,

$$A_{\lambda^*}(x) = \frac{1}{\tau}\left(\nabla_a^2 Q(x,a)|_{a=\mu(x)} + \lambda^*(x)\nabla_a^2 Q_L(x,a)|_{a=\mu(x)}\right),$$

$$\psi_{\lambda^*}(x) = -\frac{1}{\tau}\left(\nabla_a Q(x,a)|_{a=\mu(x)} + \lambda^*(x)\nabla_a Q_L(x,a)|_{a=\mu(x)}\right) - A_{\lambda^*}(x)\mu(x),$$

and

$$K_{\lambda^*}(x) = -\frac{Q(x,\mu(x)) + \lambda^*(x)Q_L(x,\mu(x))}{\tau}$$
$$+ \left(\frac{1}{\tau}\left(\nabla_a Q(x,a)|_{a=\mu(x)} + \lambda^*(x)\nabla_a Q_L(x,a)|_{a=\mu(x)}\right)^\top - \frac{1}{2}\mu(x)^\top A_{\lambda^*}(x)\right)\mu(x)$$

is a normalizing constant (that is independent of $a$). Therefore, according to the closed-form solution of the policy in (18), the optimal policy of problem (16) follows a Gaussian distribution, which is given by

$$\pi_\tau^*(\cdot|x) \sim \mathcal{N}(A_{\lambda^*}(x)^{-1}\psi_{\lambda^*}(x), A_{\lambda^*}(x)^{-1}).$$

In order to completely characterize the solution policy, it is still required to compute the Lagrange multiplier $\lambda^*(x)$, which is a polynomial root solution of (17). Since the action space is continuous, one can only approximate the integral (over actions) in this expression with numerical integration techniques, such as Gaussian quadrature, Simpson's method, or Trapezoidal rule etc. (Notice that if $\pi_B$ is a Gaussian policy, there is a tractable closed form expression for $\pi_B(\cdot|x)^\top Q_L(x,\cdot)$.)

# F  Experimental Setup

In the CMDP planning experiment, in order to demonstrate the numerical efficiency of the safe DP algorithms, we run a larger example that has a grid size of $60 \times 60$. To compute the LP policy optimization step, we use the open-source SciPy linprog solver. In terms of the computation time, on average every policy optimization iteration (over all states) in SPI and SVI takes approximately 25.0 seconds, and for this problem SVI takes around 200 iterations to converge, while SPI takes 60 iterations. On the other hand the Dual LP method computes an optimal solution, its computation time is over 9500 seconds.

In the RL experiments, we use the Adam optimizer with learning rate 0.0001. At each iteration, we collect an episode of experience (100 steps) and perform 10 training steps on batches of size 128 sampled uniformly from the replay buffer. We update the target Q networks every 10 iterations and the baseline policy every 50 iterations.

For discrete observations, we use a feed-forward neural network with hidden layers of size 16, 64, 32, and relu activations.

For image observations, we use a convolutional neural network with filters of size $3 \times 3 \times 3 \times 32$, $32 \times 3 \times 3 \times 64$, and $64 \times 3 \times 3 \times 128$, with $2 \times 2$ max-pooling and relu activations after each. We then pass the result through a 2-hidden layer network with sizes 512 and 128.

Figure 3: Results of various planning algorithms on the grid-world environment with obstacles (zoomed), with x-axis showing the obstacle density. From the leftmost column, the first figure illustrates the 2D planning domain example. The second and the third figure show the average return and the average cumulative constraint cost of the CMDP methods respectively. The fourth figure displays all the methods used in the experiment. The shaded regions indicate the 80% confidence intervals. Clearly the safe DP algorithms compute policies that are safe and have good performance.

Figure 4: Results of using a saddle-point Lagrangian optimization for solving the grid-world environment with obstacles, with x-axis in thousands of episodes.