[Reviews · NeurIPS 2018]

Reviewer 1



The focus is safe reinforcement learning under constrained markov decision process framework, where safety can be expressed as policy-dependent constraints. Two key assumptions are that (i) we have access to a safe baseline policy and (ii) this baseline policy is close enough to the unknown optimal policy under total variation distance (this is assumption 1 in the paper). A key insight into the technical approach is to augment the unknown, optimal safety constraints with some cost-shaping function, in order to turn the safety constraint into a Lyapunov function wrt the baseline policy. Similar to how identifying Lyapunov function is not trivial, this cost shaping function is also difficult to compute. So the authors propose several approximations, including solving a LP for each loop of a policy iteration and value iteration procedure. Next, based on policy distillation idea, safe policy and value iteration are extended to safe Q-learning and safe policy improvement that allow using function approximation. Experiment was done for a grid-world setting. Overall, this is a good paper. The authors provided a nice overview and comparison against previous approaches to solving Constrained MDP. But I do have some reservations about the scalability of the approach, as well as the validity of the assumptions: - the main condition of baseline policy being close to optimal is not verifiable computationally - Do we further need restricting the cost-shaping function to be a constant? (line 207) - Solving both the LP in equation (5) and policy distillation LP in equation (6) repeatedly is not scalable to large / continuous state space? (I saw from appendix F that every optimization iteration fin SPI and SVI takes 25 seconds. How about for Safe DQN and SPI?) - How would we compute the set F_L(x) (line 132-133) in order to solve the optimization in Step 1 of algorithm 1 and step 0 of algorithm 2? - Are the results reported in Figure 1 over randomized, unseen grids, or over grids used during training? Some other minor details: line 252 -> seems like some typo / D_JSD is missing The phrasing of Lemma 1 : Don't we have infinitely many possible Lyapunov function candidates? "the Lyapunov function" sounds a bit strange. The equation link in Algorithm 3 (policy distillation) is missing. Unfortunately this missing link is pretty important for understanding the algorithm. I assume this refers to equation 6. ----- After rebuttal update: I've gone through the paper again, and have read other reviews and author response. Overall I believe this is a good paper and deserves an accept. However, I'm still torn by the strong assumptions needed, the fact that the actual algorithms are still very computationally expensive to run even with the grid world domain. In addition, after reviewing the pseudocode for safe q-learning (sdqn) and safe policy improvement (sdpi) algorithms, and the grid world planning experiment, I realize that the dynamics model also needs to be known in order to update the policy in the inner loop (policy distillation step). This may further restrict the applicability of all of the proposed algorithms.

Reviewer 2



The paper focuses on the problem of safety in RL. The safety concept is expressed through a constraint on a cumulative cost. The authors leverage the framework of Constrained MDPs (CMDPs) to derive a novel approach where the Lyapunov function can be automatically constructed (instead of being manually defined). The core contribution is a method for constructing the Lyapunov function that guarantees the safety of the policy during training. In particular, safety can be efficiently expressed through a set of linear constraints. Leveraging this method, the authors showed how to modify several dynamic programming approaches to enforce safety. These algorithms do not have guarantees of convergence but the authors have empirically shown that they often learn near-optimal policies. Comments ----------- The idea proposed in the article is very interesting. In particular, I believe that safety is an important topic to be explored in RL with a potentially high impact in real-world applications. The introduction is well written and provides a nice overview of the literature about CMDPs. On contrary, I believe that preliminaries can be made clearer. I think that it is necessary to state clearly the settings. It took me a while to figure it out that the considered scenario is the stochastic shortest path problem which is an instance of total cost problem. In particular, I was surprised by the fact that the Bellman (policy) operator is a contraction mapping (that is not in general true in undiscounted settings). After checking [Bertsekas, 1995] I realized that T is a contraction since, in your problem, all stationary policies are proper [Bertsekas, 1995, Chap 2, Def 1.1, pag 80], that is, when termination is inevitable under all stationary policies. I suggest you to clearly state the settings and to put explicit references, e.g., in line 134 you should put page and prop/lem when references [7]. Am I correct? Moreover, while checking the proof in Sec C.3, I was not able to find the reference to Prop 3.3.1 and Prop 1.5.2 in [7]. Can you please provide page and or chapter? Please, also update the reference. Is it the volume 2? I believe that the analysis can be generalized outside this settings. However, you should rely on different arguments since the Bellman operator is no more a contraction in generic undiscounted problems. Could you please comment on this? Is it possible to generalize to average cost problems? Apart from that, I think that the presentation is fair and you acknowledge correctly strengths and weaknesses of the approach (e.g., the fact that algorithms do not enjoy optimality guarantees). I was surprised by the fact that there is no comparison with safe methods that guarantee a monotonic policy performance improvement: e.g., in policy iteration [Kakade and Langford, 2002; Pirotta et al, 2013a] or policy gradient [Pirotta et al, 2013b, etc.]. Your approach is definitely different but, often, the constraint cost can be incorporated in the reward (it the case of your example). The mentioned methods are guaranteed to learn policies that are always better than the baseline one. Then, if the baseline policy is good enough (satisfies the constraints reformulated in terms of bad reward) these approaches are guaranteed to recover safe policy over the entire learning process. I think you should mention these works in the paper highlighting similarities and advantages of your approach. Please comment on this. Please add a reference for the fact that the first hitting time is upper bounded by a finite quantity (you talk about standard notion). I overall like the paper and I think it will be a good fit for NIPS subject to the fact that the authors will take into account the aforementioned comments about the presentation. - Line 706: fnction -> function - Policy Distillation algorithm in page 24 contains a broken reference [Bertsekas, 1995] D. P. Bertsekas, Dynamic Programming and Optimal Control, Vol. II. Athena Scientific ISBN: 1-886529-13-2 [Pirotta et al. 2013a] M Pirotta, M Restelli, A Pecorino, D Calandriello. Safe Policy Iteration, ICML 2013 [Pirotta et al. 2013b] M Pirotta, M Restelli, L Bascetta. Adaptive step-size for policy gradient methods, NIPS 2013 ------------------------ After feedback Thank you very much for the feedback. Now it is clear to me the difference between safe approaches and the proposed algorithm. I suggest you add this explanation to the paper. Hitting time: The assumption of boundness of the first hitting time is very restrictive. I underestimated it during the review because I was not sure about the meaning of the term "uniformly bounded". I checked the reference and uniform boundness of the first hitting time means that the time to reach the absorbing state is almost surely bounded for any stationary policy. I start noticing that due to the settings you considered, every stationary policy induces an absorbing Markov chain (by definition of proper policy, [Bertsekas, 1995, Chap 2, Def 1.1, pag 80]). For absorbing Markov chains it is known that the boundness of the first hitting time (with probability 1 the time to reach the absorbing state is bounded) implies that there are no loops in the chain (every state is visited at most once with probability 1) [see for example App. D of Fruit and Lazaric, Exploration-Exploitation in MDPs with Options, 2017]. You can assume this property but I would like to see this comment in the paper because I believe that the strong implication of this assumption is not well known in general. Moreover, you need to clearly distinguish the theoretical analysis from the empirical results because this assumption seems to be not satisfied in your experiments.

Reviewer 3



This paper proposes an approach to "safe" RL where the notion of safeness comes from a constraint on another long-term expected quantity function (just like long-term cost, it could be regarded as a secondary objective function as in multi-objective RL). This approach fits in nicely with prior work on constrained MDPs. Extra care is taken in developing this safe RL algorithm so that this additional constraint is not violated at training time. This paper is very technical and can be hard to follow as a result. Some extra high-level intuition and "how things fit together" would be very useful for orienting readers. The algorithms that result from this approach (in particular algs 1 or 2) are actually pretty simple. I wonder if presenting the basic algorithms first and arguing correctness second would result in an easier reading experience. The key development of this paper is an extension to the Bellman-backup (used in value iteration for example), which accounts for the safety constraints. Developing this backup operator is not trivial (for example, the optimal policy is no longer greedy) and the analysis of it's properties requires lots of attention to detail, which the authors have taken. The end result is pretty elegant and fits into existing approaches to RL which are based on contraction mapping and fixed-point algorithms, which prior work did not do. The experimental results show promising results on a simple maze task with constraints on state-visitation frequency. Could the approach be applied to a conventional RL task like cart-pole or mountain car (with extra constraints)? If not, why? I would strongly recommend adding some discussion of scalability (runtime/sample complexity). The experiments presented are very small scale. Questions ======== - Proposition 1: What is the specific regularizer that will be added? Does it affect the solution to the optimization problem or does it only affect tie-breaking behavior? After discussion and author feedback ============================ I don't have much to add to my original review. It seems like paper #4976 is a very promising piece of work there is definitely substantial-enough contributions for a NIPS paper. However, 1) If it were my paper, I'd completely reorganize it: it's hard to disentangle the approach until you look at the pseudocode (in my opinion). The proposed algorithms are actually pretty simple, despite a dense/technical exposition of the main ideas. I am hopeful that the authors will be able to address the readability issues for camera ready. 2) As R2 explains, some of the technical assumptions required by the analysis may not be practical (see R2's discussion of bounding the hitting times and proper policies). Please be sure to add discussion of practicality and scalability for camera ready.